# Quantifying functional consequences of habitat degradation on a Caribbean coral reef

Alice E. Webb[1], Didier M. de Bakker[2,3], Karline Soetaert[4], Tamara da Costa[1], Steven M. A. C. van Heuven[5], Fleur C. van Duyl[2], Gert-Jan Reichart[1,6], Lennart J. de Nooijer[1]

[1] Department of Ocean Systems, NIOZ Royal Netherlands Institute for Sea Research, Den Hoorn, The Netherlands
[2] Department of Marine Microbiology and Biogeochemistry, NIOZ Royal Netherlands Institute for Sea Research, Den Hoorn, The Netherlands
[3] Wageningen Marine Research, Wageningen University and Research, Den Helder, The Netherlands
[4] Department of Estuarine and Delta Systems, NIOZ Royal Netherlands Institute for Sea Research, Yerseke, The Netherlands
[5] Groningen University, Faculty of Science and Engineering, Groningen, The Netherlands
[6] Department of Earth Sciences, Utrecht University, Utrecht, The Netherlands

*Correspondence to*: Alice E. Webb (webbea4@gmail.com)

**Abstract.** Coral reefs are declining worldwide. The abundance of corals has decreased alongside a rise of filter feeders, turf, and algae in response to intensifying human pressures. This shift in prevalence of functional groups alters the biogeochemical processes in tropical water ecosystems, thereby influencing reef biological functions. An urgent challenge is to understand the functional consequences of these shifts to develop suitable management strategies that aim at preserving the biological functions of reefs.

Here, we quantify biogeochemical processes supporting key reef functions (i.e., net community calcification (NCC) and production (NCP), and nutrient recycling) in situ for five different benthic assemblages currently dominating shallow degraded Caribbean reef habitats. To this end, a transparent custom-made enclosure was placed over communities dominated by either one of five functional groups: coral, turf and macroalgae, bioeroding sponges, cyanobacterial mats, or sand, to determine chemical fluxes between these communities and the overlying water, during both day and night. To account for the simultaneous influence that distinct biogeochemical processes have on measured variables, the rates were then derived by solving a model consisting of differential equations describing the contribution of each process to the measured chemical fluxes.

Inferred rates were low compared to those known for reef flats worldwide. Reduced accretion potential was recorded, with negative or very modest net community calcification rates by all communities. Net production during the day was also low, suggesting limited accumulation of biomass through photosynthesis and remineralisation of organic matter at night was relatively high in comparison, resulting in net heterotrophy over the survey period by most communities. Estimated recycling through nitrification and denitrification were high but denitrification did not fully counterbalance nutrient release from aerobic mineralisation, rendering all substrates sources of nitrogen. Results suggest similar directions and magnitudes of key biogeochemical processes of distinct communities on this shallow Curaçaon reef. We infer that the amount and type of organic matter released by abundant algal turfs and cyanobacterial mats on this reef likely enhances heterotroph activity and stimulates

the proliferation of less diverse copiotrophic microbial populations, rendering the studied reef net heterotrophic and drawing
the biogeochemical 'behaviour' of distinct communities closer to each other.

## 1 Introduction

Community composition and biodiversity across all kinds of ecosystems are responding to escalating anthropogenic activities (McGill et al. 2015). In both terrestrial and aquatic systems, climate change, pollution and habitat fragmentation have promoted the expansion of opportunistic and tolerant species and the elimination of more sensitive yet key specialists (Clavel et al. 2011). Communities within ecosystems and across spatial scales have become more biologically homogeneous (Burman et al. 2012; Cramer et al. 2021) which may lead to a decrease in functional diversity, therefore limiting services provided by biological communities (Matsuzaki et al. 2013; White et al. 2018). Additionally, this may cause synchronisation of the biological response to new or intensified anthropogenic pressures across local communities, thus reducing resilience of metacommunities (Tobias and Monika, 2012; Sonnier et al. 2014; Petsch et al. 2020).

Coral reefs support immense biodiversity and provide important ecosystem services to millions of people (Moberg and Folke, 1999). They are, however, in global decline as they are experiencing major loss in coral abundance and shifts in species composition in response to increasing human pressures and accelerating rates of environmental and climate change (Koop et al. 2001; Langdon and Atkinson, 2005; Andersson and Gledhill, 2013; De'ath et al. 2012; Chen et al. 2015). Returning degraded reefs to their original state is, in many cases, no longer an option (Hughes et al. 2017). Instead, today's challenge is to guide coral reefs through this transition while identifying and securing the ecosystem functions that underpin resilience and services of modern reef assemblages (Oliver et al. 2015).

This is particularly relevant for the depauperate reef systems in the Caribbean, which have, since the early 1970s, undergone considerable reorganisation with regards to community composition and structural appearance (Gardner et al. 2003; Jackson et al. 2014). The communities encountered on these reefs bear little resemblance to the systems once dominated by reef-building *Acropora* spp. and *Orbicella* spp. (van Duyl, 1985; Alvarez-Filip et al. 2009). Major declines in the abundance of these species severely compromise reef function as they were the main drivers of critical processes including carbonate accretion, productivity, and structural complexity (Wolfe et al. 2020). This has led to low functional redundancy on Caribbean reefs, i.e., a reduced capacity of one or more species to functionally compensate for the loss of another, which makes them particularly vulnerable (Bellwood et al. 2003; McWilliam et al., 2018). On many reefs, areas covered by turf assemblages and macroalgae, excavating sponges, cyanobacteria, rubble, and sand have increased alongside the decrease in stony corals (Aronson et al. 2005; Burman et al. 2012; Cramer et al. 2021). Although changes in community composition are well documented as they can be followed by monitoring the coverage of the various benthic taxa over time (Barott et al. 2012; de Bakker et al. 2016; de Bakker et al. 2017), assessing the impact of these shifts on the community ecophysiology in situ has proven more challenging.

The keystones of coral reef functioning include provision of a structural habitat through carbonate deposition, production and assimilation of biomass produced through photosynthesis and efficient cycling of nutrients within the ecosystem (Brandl et al. 2019). The biogeochemical processes that underlie these key functions are primary production, aerobic mineralisation, calcification, bioerosion, and nutrient release/uptake. Complementary to conventional monitoring efforts, quantification of the net budgets of these processes will provide insight into how reef degradation and community reorganisation affect reef functioning (Brandl et al. 2019, Bellwood et al. 2019). However, obtaining accurate in situ measurements while accounting for the complexity of interactions between processes can render their quantification rather complicated.

In environments where the flow of water over the reef is relatively linear, the upstream/downstream method can be performed (Shaw et al., 2014; Koweek et al., 2015; Albright et al., 2016, 2018). For flow regimes that are not unidirectional, factors such as water residence time and biochemical and hydrological offshore conditions need to be considered (Courtney et al., 2016). When conditions allow build-up of considerable chemical vertical gradients, net fluxes (of e.g., nutrients) can be measured (McGillis et al., 2011; Takeshita et al., 2016). On fully exposed reefs, where virtually no detectable accumulation occurs over the reef-flat - even within the boundary layer - incubating communities allow quantification of the fluxes into and out of the overlying water. Presently, efforts to quantify community processes have focused on individual functional groups (Brocke et al. 2015, 2018; den Haan et al. 2016; Webb et al. 2017; de Bakker et al. 2018) or on reconstructed communities ex situ (e.g., Dove et al. 2013, 2020). Moreover, the few studies that incubated whole communities in situ have so far not accounted for the complexity of interactions between biogeochemical processes (Yates and Halley, 2003, Kline et al. 2012, van Heuven et al. 2018, Roth et al. 2020, 2021).

Here, biogeochemical processes underlying key reef functions were quantified in situ across five different benthic assemblages found on the fringing reef of Curaçao, consisting of functional groups that currently characterise many degraded shallow reef habitats throughout the wider Caribbean. To this end, a custom-made tent was placed over substrates dominated by either 1) coral, 2) turf and macroalgae, 3) bioeroding sponges, 4) benthic cyanobacteria mats or 5) sand. Chemical fluxes between water column and reef were then determined by monitoring nutrients, inorganic carbon chemistry and oxygen. This was done both during the day and at night to estimate overall net metabolism of these communities. To account for the simultaneous convoluted influence that various processes have on measured variables, the change in their concentrations is related to the responsible metabolic processes by solving a model consisting of ordinary differential equations describing the contribution of each process to the measured chemical fluxes. With this approach, model parameters (i.e., rates of biogeochemical processes) are derived from concurrent changes in all measured variables. The aim being to provide accurate estimates of the rates of the biogeochemical processes that underlie functions of the newly configured shallow Caribbean reefs.

## 2 Materials and Methods

### 2.1 Study Site

Reef incubations were carried out on the leeward side of Curaçao (Piscadera Bay; 12°07'16.3"N 68°58'13.2"W) between the 12th of February and 22nd of March 2018, at depths ranging from 5 to 7 meters. The water at the study site is characterised by episodes of high turbidity and is periodically eutrophied due to terrestrial runoff and ineffective waste-water treatment. Sediment plumes transporting high concentrations of nitrate, ammonium and phosphate into the shore's fringing reef are commonly encountered after a period of heavy rainfall (den Haan et al. 2016). The shallow reef flat nearby the entrance of the bay in which we conducted our incubations is characterised by rubble and patchy distribution of small coral heads making this location particularly suitable for the deployment of tent incubations.

### 2.2 Tent Incubations

The incubation enclosure consists of a custom made, tetrahedron-shaped "tent" (Fig. 1). It has transparent, vinyl-and-butanyl walls with rigid pole edges of 1 m in length, resembling the cBIT described by Haas et al. (2013). It also includes 0.5 m long flaps extending outward from each of the tent's three sides, allowing for better sealing of the tent to the substrate by placing weights (metal chains) on these flaps. The enclosure covers a 0.43 m$^2$ planar surface and encloses a 118 L volume. All three sides of the tent contained an opening to allow flushing of the enclosed volume between incubations: during incubations these openings were sealed by zippers. Water enclosed in the incubation tent was homogenised during the experiment by means of a continuously running brushless submergible water pump (BLDC pump Co., Ltd.). This pump was attached to one of the tent poles, at half the height of the tent, generating a vertical circulating turbulence, while minimising the upsurge of sediment. Effectiveness of the stirring was demonstrated by rapid and even dispersal of a small dose of injected fluorescein prior to the incubation. Surge movement was retained due to the non-rigid texture of the tent walls. Incubated communities included five different types of substrate dominated either by turf and macroalgae, sand, bioeroding sponges, benthic cyanobacteria mats or coral (Fig. 2), equalling a total of 15 studied communities (three of each type). Each community was incubated during the day (n=15) and due to practical reasons, only 2 of each type were incubated during the night (n=10) (i.e., for each type of community, three daytime and 2 night-time incubations were carried out).

The incubations were carried out one at a time, over the study period and lasted four hours each. Prior to each incubation, the tent was placed with flaps open over the substrate and lefts for a minimum of 3 hours before the incubation was started. When day incubations were terminated, the tent was left in place with flaps open so that night incubation coulb be carried out on the same substrate. Daytime incubations were started at 10:00 and night-time incubations started at 18:30.

### 2.3 Substrate Compositions

Substrates dominated by either coral, turf and macroalgae (TMA), bioeroding sponges (BES), cyanobacteria mats (BCM), or sand were incubated (Fig. 2). Three reef patches of each reef assemblage type were chosen depending on their cover of the

dominant benthic component (see Table S1 for detailed species composition and cover). In some cases, to fit adequate

incubation location and the tent capacity, pieces of rubble infested with sponge or covered in turf were added or retrieved from

the community. In these cases, the community was left to stabilise two or three days before starting incubations. Incubated

substrate included colonised hard substrate surrounded by bare hard substrate covered in a fine layer of sand for better enclosure

deployment (except for sand incubations). The incubated coral species are characteristic of degraded Caribbean reefs and

include some of the most prominent tolerant and opportunistic species found on modern reefs (Darling et al. 2012; de Bakker

et al. 2016; Cramer et al. 2021) (see Table S1). Turf here refers to the epilithic algal matrix, defined by Clements et al. (2016)

as 'a conglomeration of short, turf-forming filamentous algae (< 1 cm high), macroalgal spores, microalgae, sediment, detritus

and associated fauna'. The benthic cyanobacterial mats in all three tent replicates were thick brown/reddish in colour and in

line with the description in Brocke et al. (2018) for mats found between 3 and 7 meters dominated by the species *Oscillatoria*

*bonnemaisonii*. Percentage cover was measured in situ after removal of the tent. For substrates dominated by coral, its cover

ranged from 34 to 36%. Turf and macroalgae cover ranged between 72 to 83%, bioeroding sponge cover varied from 38 to

40%, and cyanobacterial mats cover ranged from 83 to 91%, in their respective incubations (Fig. 2).

### 2.4 *In Situ* Measurements

Measurements of salinity (S), temperature (T), dissolved oxygen ($O_2$) and photosynthetically active radiation (PAR) within

the tent were recorded at one-minute intervals throughout the duration of the incubations. S and T were measured using a Star-

Oddi DST CTD, $O_2$ was recorded using a HOBO U26 dissolved oxygen sensor and data logger and PAR was assessed by an

Odyssey light logger (Dataflow Systems PTY Ltd., Christchurch, NZ), calibrated in air against a Walz instrument (Walz

ULM500, Walz GmbH, Effeltrich, Germany). In addition, S, T and PAR were measured for the duration of the incubations

outside the tent using the same sampling frequency. All instruments within the tent were attached to the ridges except the

Odyssey logger which was placed on the substrate facing upwards (covering approximately 150 $cm^2$ of the substrate).

### 2.5 Discrete Sampling

During each incubation, discrete samples were collected both inside and outside the tent at $T_0$, after two ($T_2$) and after four

hours ($T_4$) by SCUBA diving for the analyses of total alkalinity ($A_T$), total inorganic carbon ($C_T$) and nutrients ($NO_2 + NO_3$,

$NO_2$ and $NH_4$). pH was calculated from the former two parameters using the package Seacarb (Lavigne et al. 2009). Sampling

of the tent interior was carried out from the outside by drawing seawater through 150 ml plastic syringes connected to a 1.5 m

gas-impermeable tube (Tygon; Fig. 1). Syringes were flushed three times with the sampling water before collecting an actual

sample. The tubing was fixed around a rigid pole of the tent in such way that the seawater was sampled from the centre of the

tent incubation. The tube end located inside the tent was equipped with a Whatman ® filter (G/F 0.47 µm) which was replaced

daily to avoid the collection of particulate matter.

Analyses for $A_T$ were performed within two hours upon sampling using spectrophotometrically guided single-step acid titration

(Liu et al. 2015) and samples for $C_T$ were run on an autoanalyser Traacs 800 spectrophotometric system (Stoll et al. 2001) at

the NIOZ (Royal Netherlands institute for sea research). Accuracy of both instruments was set using certified reference material supplied by Scripps Institute of Oceanography (Dickson et al. 2007). Precision of replicates was 2.7 µmol kg$^{-1}$ for $C_T$ and 0.9 µmol kg$^{-1}$ for $A_T$. Samples for dissolved inorganic macronutrients were prepared by dispensing sampled water through 0.8/0.2 µm Acrodisk filters into 5 mL pony vials, and subsequently stored at -20 ºC until analysis at the NIOZ on a QuAAtro continuous flow analyser (SEAL Analytical, GmbH, Norderstedt, Germany) following GO-SHIP protocol (Hydes et al. 2010).

## 2.6 Rates of Water Exchange

After sampling water at $T_0$ for $A_T$, $C_T$ and nutrients in each incubation, 450ml of salt-saturated water was injected into the tent. The rate at which the elevated interior salinity equilibrated with ambient salinity during the incubation was used to estimate the rate of water exchange with the surrounding sea water for each incubation.

The rate of change of salinity within the incubation can be solved by the differential equation below:

$$\frac{dS}{dt} = K\ (Sout - S)$$

Where $\frac{dS}{dt}$ is the rate at which salinity changes within the tent, Sout is the exterior salinity, S is the interior salinity and K is the water exchange rate.

The equation is solved using the function 'ode', within the package deSolve (Soetaert et al. 2010), the R routine that solves the differential equations. Function 'modfit' from the package FME (Soetaert and Petzoldt, 2010) was used to perform iterative minimisation (based on least squares) on residuals to find the best fit within lower and upper bounds.

## 2.7 Inverse Modelling and Model-Data Comparison

The use of inverse modelling is advantageous as it enables us to derive unknown parameters (here rates of biogeochemical processes) simultaneously from all measured data. The mathematical "state" of the incubation's dynamic system can be described based on the mass balance between $A_T$, DIC, $O_2$, $NH_4$ and $NO_3$ which is influenced by various biogeochemical processes. The rate of these processes are the unknown parameters that need to be quantified by fitting against an incomplete data set (only three time-points for $A_T$, DIC, $NH_4$ and $NO_3$).

The model consists of the five differential equations depicted below that relate the change in measured variable concentrations over time to the responsible processes, which are here assumed to have remained constant over time.

Since the involved processes affect the different chemical components simultaneously, the combination of these differential equations can be used to solve the contribution (in terms of rates) of the processes to the observed changes. The processes in question include aerobic mineralisation ($O_2$ consumption related to mineralisation), primary production (PP), calcification, dissolution, nitrification and denitrification.

$$\frac{dC_T}{dt} = mineralisation - PP - calcification + dissolution + K\,(C_T out - C_T)$$

$$\frac{dNH_4}{dt} = mineralisation \times NCratio - PP \times NCratio \times Pnh4 - nitrification + K\,(NH_4 out - NH_4)$$

$$\frac{dNO_3}{dt} = -(mineralisation \times pDeni \times 0.8) - PP \times NC_{ratio} \times (1 - Pnh4) + nitrification + K\,(NO_3 out - NO_3)$$

$$\frac{dA_T}{dt} = -2 \times calcification + 2 \times dissolution + \frac{dNH_4}{dt} - \frac{dNO_3}{dt} + K\,(A_T out - A_T)$$

$$\frac{dO_2}{dt} = -mineralisation \times OC_{ratio} \times (1 - pDeni) + PP \times OC_{ratio} - 2 \times nitrification + K\,(O_2 meanout - O_2)$$

With mineralisation describing the degradation of an organic compound to its mineral components, i.e. carbon dioxide and inorganic nutrients. PP is the primary production and calcification is the deposition of calcium carbonate. Pnh4 is the part of
195 N uptake as $NH_4$ for primary production. Dissolution results in an increase of calcium and carbonate ions by degradation of calcium carbonate shells and/or skeletons and K is the water exchange rate. Nitrification is the process by which ammonium ($NH_4^+$) is converted into nitrate ($NO_3^-$); two moles of oxygen are needed to oxidize one mole of ammonium during nitrification. pDeni is the fraction of mineralisation that respires nitrate (i.e., denitrification). The $OC_{ratio}$ is the ratio between the concentrations of oxygen and $C_T$. The $NC_{ratio}$ is the ratio between N and $C_T$. The 0.8 constant refers to the denitrification redox
reaction (Soetaert et al. 2007).

We start by determining the parameters that can be fitted, based on parameter collinearity. After producing a best-fit set of the selected parameters, we quantify parameter uncertainty, and produce sensitivity ranges around the modelled variables.

The $OC_{ratio}$, $NC_{ratio}$ and K parameters are always fixed and estimated from data prior to running the model. Others vary between fixed and free (to be fitted) depending on collinearity and light. For instance, primary production is fixed at 0 during night
incubation, however during the day, only the dominant process can be estimated. Some parameters are highly correlated with each other such as primary production and remineralisation or calcification and dissolution and therefore cannot be estimated simultaneously. In general, when the collinearity index exceeds 20, the linear dependence is assumed to be critical (i.e., it will be impossible or difficult to estimate all the parameters in the combination together).

Collinearity of the parameter sets is measured using function 'collin' within the FME package (Soetaert and Petzold, 2010).
The model equations are specified in a function that calculates the rate of change of the state variables ($dC_T$, $dNH_4$, etc). Inputs to the function are the model time (t), the values of the state variables ($C_T$, $NH_4$, $NO_3$, $A_T$ and $O_2$) and the parameters (remineralisation, calcification, etc.). The differential equation model is solved using function 'ode', within the package deSolve, the R routine that solves the differential equations.

The discrepancy of the model solution with observed changes within the tents is calculated using function 'modCost' still in
the FME package which estimates the residuals and the variable and model costs (sum of squared residuals).

Function 'modfit' was then used to perform iterative minimisation (based on least squares) on residuals to find the parameter giving the best fit within lower and upper bounds. Estimated parameters are the unknown fluxes (mineralisation, PP, calcification, etc.).

## 2.8 Conversion to fluxes

The best-fit parameters, i.e., the input rates R (in $\mu mol\ kg^{-1}\ min^{-1}$), in the tent are converted to fluxes from the water-substrate interface ($mmol\ m^{-2}\ h^{-1}$), assuming an enclosed mass of water of $108 \pm 10$ kg (tent encloses approximately 118 litres of volume; of which substrate volume is ~10 L; seawater density ~1022 kg m$^{-3}$) and an incubated planar surface of 0.43 m$^2$. Net community calcification (NCC) fluxes were determined from the predicted calcification and dissolution. The model captures the dominant net flux and does not distinguish the relative contributions of gross calcification and dissolution to the integrated NCC rate. Net community production (NCP) is the difference between remineralisation and primary production. Denitrification is estimated from the pDeni fraction and the mineralisation parameters.

## 2.9 Comparing biogeochemical signatures between incubations

To evaluate if water exchange rate had an impact on estimated processes, the non-parametric Kendall rank correlation test was performed. All inferred biogeochemical process rates (mineralisation, primary production, NCP, NCC, nitrification and denitrification) were tested against incubation water exchange rates.

Principal component analysis (PCA) was used to identify grouping among the 23 tent incubations (day n = 13, night: n = 10) in relation to their biogeochemical signature (i.e., NCC, NCP, nitrification and denitrification). The oxygen logger malfunctioned during two of the day incubations consisting of one BCM and one TMA- dominated community incubation. The model was therefore not run through these tents. PCA was conducted on a centred multivariate data set consisting of the four main biogeochemical processes (i.e., NCC, NCP, nitrification and denitrification). Additionally, NCC was plotted against NCP to evaluate how the balance between both processes varied among distinct communities and by which process were communities dominated.

## 3. Results

### 3.1 Ambient conditions

In-tent light and temperature were only slightly impacted by the tent enclosure compared to the exterior (Fig. 3). Light was on average 17 % lower inside the tent and changes in temperature were dampened within the tent. In-tent temperatures were on average 0.2 °C higher than those outside the tent. Average ambient $A_T$, DIC, pH, NH$_4$ and NO$_3$ was $2386.8 \pm 13.9$, $2125.5 \pm 20.0$, $7.9 \pm 0.003$, $0.31 \pm 0.15$ and $0.32 \pm 0.14$ $\mu mol\ kg^{-1}$ respectively. Measured data for each incubation, inside and outside the tent for all three time-points, as well as the differences between $T_0$ and $T_4$ are presented in Table S2.

## 3.2 Water exchange quantification

Application of Equation (1) to salinity data collected during all incubations yielded dilution rate K ranging between 0.004 and 0.044 min$^{-1}$. This indicates that 2.1 to 4.8 kg of seawater (i.e., $K \times 108$ kg) was exchanged every minute between the incubation enclosure and the environment. These rates correspond to the intensity of the water movement observed and recorded visually at the time of each incubation. Fig. 4 shows the data used to estimate the rate of water exchange of an incubation with relatively minor leakage (A) and one where leakage is more severe (B). In these examples, in-tent salinity returns to ambient concentrations after ~1 and ~2 hours respectively.

## 3.3 Model output

Figure 5 illustrates the output of our approach for all incubations. Using a minimisation routine, best fit parameters (mineralisation, PP, calcification, etc.) were predicted to best fit the model to the measured data. Individual graphic output for two incubations (including one carried out during the day on substrate dominated by BES and one performed at night-time on BCM-dominated substrate) with respective fixed and fitted parameters are presented in Fig. S1. Details for parameter prediction and best-fit can be found in Table S3. The model output shows a relatively good fit to the measured observations (Fig. 5), indicating that the interactions between processes and their effects on chemical fluxes were considered correctly. Overall fit is usually better on night data, which is mostly due to the inability of the model to predict irregular oxygen evolution caused by light variability during the daytime (Table S3).

As the process estimates are limited to net increase or decrease, fluxes for PP and mineralisation are presented as net community production (NCP) and calcification and dissolution are combined into net community calcification (NCC; Fig. 6).

## 3.4 Estimated biogeochemical processes

NCP showed a clear diurnal pattern (Fig. 6). While all NCP values were modestly skewed towards net autotrophy during the day (except for sand), the strongest signal was found for substrates dominated by BCM with an average daily NCP of 5.6 mmol m$^{-2}$ h$^{-1}$. Night values indicated net respiration ranging from an average of -2.64 mmol m$^{-2}$ h$^{-1}$ on substrates dominated by TMA to -16.28 mmol m$^{-2}$ h$^{-1}$ on substrates dominated by bioeroding sponges.

A clear diurnal signal also resided in NCC fluxes for all substrates involved (Fig. 6). Most NCC fluxes recorded during daytime (except for sand incubations) indicated net CaCO$_3$ precipitation. At night, most NCC fluxes indicated net CaCO$_3$ dissolution, especially on substrates dominated by BES and BCM. The absence of change in A$_T$ for coral dominated substrates during the night indicated that dissolution equalled calcification during these incubations and hence, the average NCC at night was close to 0. Substrates dominated by coral generated the strongest decrease in A$_T$ (net precipitation) during daytime, yielding an average NCC rates of 0.45 mmol CaCO$_3$ m$^{-2}$ h$^{-1}$. Highest net dissolution was found at night-time for incubations of substrates

dominated by bioeroding sponges and cyanobacterial mats, with a comparable average of 0.56 and 0.63 mmol $CaCO_3$ $m^{-2}$ $h^{-1}$ respectively.

Nitrification was found to occur predominantly at night with higher fluxes in incubations of substrates dominated by bioeroding sponges, cyanobacterial mats and corals. Denitrification also occurred mostly at night except on sand where daytime and night-time fluxes were small but relatively similar (Fig. 6).

The Kendall rank correlation test did not reveal significant correlation between water exchange rates and rates of mineralisation (p=0.79, tau=0.04), primary production (p=0.47, tau=0.12), NCP (p=0.75, tau=0.05), NCC (p=0.17, tau=0.21). Nitrification (p=0.81, tau=0.04), and denitrification (p=0.27, tau=0.18). The Kendall correlation coefficient tau is closer to zero than to one in all cases, implying there is no significant association between the two tested variables.

### 3.5 Incubation comparison

The PCA based on the four main biogeochemical processes revealed different groups for night incubations and day incubations (Figure 7A), except for sand incubations where night and day incubations grouped relatively close to each other. The first two principal component axes (PC1 and PC2) explained 88.68% of the total variability within the data. PC1 described a gradient in NCP and NCC from high (negative PCA scores) to low (positive PCA scores) and an opposite pattern for nitrification and denitrification. PC2 further explains the variability in NCC and nitrification, and to a lesser extent NCP and denitrification. One of the communities dominated by bioeroding sponges (rep.1) is separate from other communities both during the day and during the night due to considerably higher rates of nitrification and denitrification compared to other communities.

Figure 7B showed the position of the 15 communities in the different quadrants of the NCC vs. NCP diagram. A clear separation is observed between day incubations, characterised by net photosynthesis and calcification and night incubation indicating net respiration and dissolution. Day-time sand incubations are centred due to the low magnitude of the processes occurring on these substrates and are the only incubations exhibiting net dissolution during the daytime. For night-time incubations, only coral-dominated communities displayed modest net calcification.

## 4. Discussion

The biogeochemical flux assessment has enabled us to identify and quantify the biological functions that are currently at play on this degraded Curaçaon reef flat. Although the present study only investigated the shallow part of one single reef, it provides insight into the effects that shifts in coral species and functional groups has had on the overall functioning of these communities. Comparison with coral reefs in a broader biogeographical context is needed to establish whether the rates obtained here are site-specific or representative of degrading Caribbean coral reefs in general.

The shallow reef communities investigated at this site barely support reef functions that are usually ascribed to a healthy coral reef. Overall, net community calcification and production on these substrates were low compared to reef flats worldwide (Atkinson, 2011). Very low or negative NCC rates were recorded on all substrates, suggesting reduced net accretion potential. Net production was also low, likely indicating limited accumulation of biomass, while heterotrophic processes were prominent. Recycling processes, nitrification and denitrification, were high but did not prevent net nutrient release from aerobic mineralisation, rendering all substrates, sources of nitrogen. Although processes recorded on substrates dominated by coral, bioeroding sponges and cyanobacterial mats showed some variation between types of substrates, the overall behaviour of complementary processes for each of these assemblages was relatively comparable in terms of direction and magnitude.

## 4.1 Net Dissolving Reef

Although net calcification was recorded during the day on all substrate types (except sand), it did not compensate for higher dissolution rates at night except on substrates dominated by coral. Diel shifts between net calcification and net dissolution are usual for coral reefs and have been recorded on healthier systems than the one studied here (Yates and Halley, 2003; Albright et al. 2013; Albright et al. 2015; Koweek et al. 2015) with instances of net dissolution mainly taking place at night, coinciding with net respiration (and most likely with low gross calcification) (Cyronak et al. 2018). However, the community calcification budget over 24 hours resulting from this diurnal variability in the present study resulted in a modest average net calcification rate of 5.7 mmol $CaCO_3$ m$^{-2}$ day$^{-1}$ on coral-dominated substrates. This is low compared to rates reported for reef flats worldwide (with an average around 130 mmol $CaCO_3$ m$^{-2}$ day$^{-1}$ and ranging from 20 to 250 mmol $CaCO_3$ m$^{-2}$ day$^{-1}$; Atkinson, 2011). Overall, the limited number of in situ flux-based experiments carried out in the wider Caribbean (Yates and Halley 2003; Muehllehner et al. 2016; van Heuven et al. 2018) suggest they are among the lowest NCC rates recorded worldwide (Albright et al. 2015; Shaw et al. 2015; Silverman, 2007) and particularly low compared to those in the Indo-Pacific region (Koweek et al. 2015; Takeshita et al. 2016). Surveys using a census-based approach (Perry et al. 2013; de Bakker et al. 2019) also showed that some Caribbean reefs are net eroding. The recorded low rates of net carbonate production in the wider Caribbean may be expected simply due to the region-wide decrease in coral coverage since the 1970s. However, the decrease in net calcification relative to historical values is likely related to more than just coral cover loss. Indeed, the latter has subsequently left surfaces available for colonisation by turf, macroalgae (Hughes, 1994) and more recently, cyanobacterial mats (de Bakker et al. 2017). Shallow reefs around Curaçao (<10 m) are covered by filamentous algal turf canopies that presently represent the most dominant benthic component on these reefs (Vermeij et al. 2010). Given their abundance and high release rates of dissolved organic carbon (Mueller et al. 2016), heterotrophic activity is likely to be stimulated. Furthermore, cyanobacterial mats release part of their photosynthetically fixed carbon as DOC into the water column at a higher rate than turf and macroalgae (Mueller et al. 2014; Brocke et al. 2015). They have been shown to be responsible for 79% of the total DOM release over a 24 h diel cycle at this same study site (Brocke et al. 2015). Considering their proliferation around the islands of Curaçao and Bonaire since ~2003 (de Bakker et al. 2017) and the prevalence of turf algae in the area, an accumulation of organic matter may have resulted in a reduction of pH due to oxidation of organic matter, i.e. stimulated heterotrophic activity, resulting in reduced

calcification (Bates et al. 2010). Muehllehner et al. (2016) suggested that the seasonal character of reef dissolution they recorded on the Florida Reef Tract coincided with an accumulation of organic matter, following the die-off of annual sea grasses in the area.

### 4.2 Net heterotrophic reef

Low net community production rates in the current study indicate that autotrophic processes dominate modestly during the day. Integrating the NCP values over 24 hours (day + night) yielded rates skewed towards net respiration, indicating heterotrophy in all incubations. Although net community production of reef flats has been reported to vary notably over the course of the day (Koweek et al. 2015), with values ranging from -220 to +310 mmol m$^{-2}$ day$^{-1}$ (Atkinson, 2011), large amplitude shifts between net autotrophy and net heterotrophy are usually recorded between day and night (Yates and Halley,
2003; Albright et al. 2013; Albright et al. 2015; Koweek et al. 2015). Here, the amplitude of this shift between day and night was modest. It should be noted however, that the reduction in light intensity by 17 % on average may have resulted in a slight underestimation of NCP rates during the daytime. This would hold especially true for BCM incubations. Reductions in light would intensify down the steep vertical physiochemical gradients present in these microbial mats and could interfere with light-controlled circadian regulation of photosynthesis and respiration in these cooperative communities (Hörnlein et al. 2018),
favouring respiration and decreasing net community productivity.

The reduction in the amplitude of the diel shift in net production and calcification recorded in the present study may have severe implications. For instance, metabolic fluctuations from reef biota cause strong temporal fluxes in compounds which affect the oscillatory behaviour of reef seawater microbial communities (Kelly et al. 2019; Weber et al. 2020) leading to less distinct populations and more redundancy in microbial specialists' functions, i.e., a shift to a dominance in catabolic pathways.
Organic material supplied to the ecosystem by benthic primary producers as exudates is also thought to play a pivotal role on microbial growth (Haas et al. 2011) and diversity (Nelson et al. 2013; Haas et al. 2016) depending on its origin. Studies on the effect of exudates of macroalgae and turf on microbial metabolism demonstrated that the composition of exudates stimulated rapid growth of less diverse microbe communities compared to coral derived exudates. Consequently, reef microbial communities shift towards copiotrophic populations that have the potential to remineralise available organic nutrients at a high
rate and encode greater numbers of potential virulence factor genes, ultimately harming corals and maintaining algal dominance (Nelson et al. 2013; Dinsdale and Rohwer, 2011). We infer that the amount and type of organic matter provided by abundant algal turfs mats on this reef, likely enhances heterotroph activity and stimulates the proliferation of less diverse copiotrophic microbial populations, rendering the studied reef net heterotrophic regardless of substrate type.

### 4.3 Nitrogen cycling

Nitrogen pathways support high primary productivity in oligotrophic environments by supplying nutrients while simultaneously preventing the build-up of excess nutrients that may favour opportunistic primary producers such as algal turfs

(e.g., O'Neil and Capone, 2008; Karcher et al. 2020). The abundance of non-coral primary producers on these reefs suggest that nitrogen is not a limiting factor for growth. Results showed that all substrate types acted as $NH_4^+$ and $NO_3^-$ sources during the day and the night, apart from sand and turf substrates which acted as sinks for $NO_3^-$. This is to be expected from overall net heterotrophic communities, however, even in instances of net autotrophy during the day, substrates still acted as DIN sources. This is comparable to recent results from in situ incubations carried out in the central Red Sea on net autotrophic coral and algae-dominated communities (Roth et al. 2020, Roth and 2021). It is likely that other community-wide processes, such as the consumption and transformation of organic matter by microbial populations (e.g., Pfister and Altabet, 2019), masked the assimilation of DIN by primary producers.

Nitrification and denitrification rates measured in the present study generally fall within the published range of in situ measurements in tidal pools dominated by algae and corals (Webb and Wieber, 1975), in cavities covered in encrusting sponges (Scheffers et al. 2004), on cyanobacterial mats (Bonin and Michotey, 2006) and on carbonate sand (Capone et al. 1992; Eyre et al. 2013b). However, there was no nitrification during the day (except for one community dominated by sponges), which may be explained by light causing a reduced activity of nitrifiers (Kwon et al. 2020). Owing to the rather shallow depths of our experiment, nitrifiers may have been negatively affected by the light. As mentioned above, microbial communities are impacted by organic matter composition and temporal fluctuations in biochemicals. Shifts in diversity and abundance of the microbial communities inhabiting the reef substrate may also lead to diel shifts in nitrogen-cycling capacity (Rädecker et al. 2015). Further research investigating how alterations in diversity and abundance of these microbial functional groups relate to changes in the nitrogen-cycling capacity of reef assemblages is needed at this point.

**4.4 Similar biogeochemical processes by reef communities dominated by corals, sponges, and cyanobacteria**

Although processes recorded on distinct community assemblages showed some variation between substrate types, present results suggest that the various communities of this degraded reef of Curaçao exhibit similar directions and magnitudes of key net biogeochemical processes. Results indicate that even on substrates with coral cover ranging from 34% to 36%, which is high for Curaçaon reefs and relative to the wider Caribbean region (Jackson et al. 2014; de Bakker et al. 2016 and 2019), net community calcification is very low. In fact, daily rates are in a similar range to those for substrates dominated by bioeroding sponges where coral cover ranged from 1 to 9% and to substrates covered by cyanobacterial mats where no live coral was recorded (coral: 0.45, BES: 0.45 and BCM: 0.33 mmol $CaCO_3$ m$^{-2}$ h$^{-1}$). Recent work by Romanó de Orte et al. (2021) found comparable results showing similar daytime calcification rates for live coral and dead coral substrate. Although hard coral is generally assumed to dominate the calcification signal on tropical reefs, these results suggest that coral might not be the sole key player in coral reef calcification dynamics on such impacted sites. Cementation/lithification processes carried out by coralline calcifying algae, micro-calcifiers (e.g., foraminifera and juvenile shells) and benthic microbial communities, resulting in the trapping and binding of rubble and sediment in cryptic habitats and within/on the rubble, may play a comparably important role, counteracting some of the dissolution occurring in these communities.

The main differences between coral-dominated substrates and others, in terms of NCC, is that they were the only substrate able to balance out night-time dissolution. Primary production is barely compensating heterotrophic processes during the day on all substrates. Although substrates incubated in the present study are distinct in taxa dominance, they do share some similarities that may be drawing biogeochemical process differences closer to each other regardless of substrate type. For instance, turf covers any part of hard substrate available, the sand and rubble potentially harbour a variety of comparable cryptic organisms and the microbial community within and above each of these substrates may be shifting similarly towards generalist copiotrophic populations.

Shifts in community composition have resulted in the impairment of key reef processes and the present results may suggest that some degree of functional homogenisation (Clavel et al. 2011) exists among substrates with different community assemblages. It is noteworthy that seasonality may explain biogeochemical process similarity between major biogeochemical processes on reefs dominated by distinct functional groups. Roth et al. (2020) recently found that summer temperatures amplified functional differences between coral- and algae-dominated communities in the central Red Sea. Higher temperatures benefit algae-dominated communities in terms of primary production and growth while coral-dominated communities shifted towards a more heterotrophic state with depressed net community calcification rates. The fact that coral-dominated substrates studied here are already in a heterotrophic state with very low NCC values in winter temperatures attests to the differences in the studied systems and provides an opportunity for comparison between a relatively healthy system and a degraded one (Roff and Mumby, 2012). Additionally, average ambient pH at the current study site was 7.9, which is lower than average 'summer' pH, usually between 8.1 and 8.2 (den Haan et al. 2016). This may suggest that depressed calcification rates in the Piscadera Bay are indeed linked to seasonality. However, further research and additional incubations are needed to better understand the effect of seasonal variation on the functional states of these degraded reefs.

## 4.5 Method considerations

The combination of in situ incubations and inverse modelling, which incorporates the complexity of interactions between processes, has proven to be an effective tool to provide quantitative data on the functional state of coral reef patches. Quantification of the exchange of substances between reef communities and the overlaying water was achieved despite the presence of swell-induced seawater exchange because this approach allowed for a volume exchange between the environment and the incubation. The incubations can thereby be replenished at some extent, keeping saturated $O_2$ levels within the tent, thus minimising unrepresentative reef community metabolism. Fluxes within the incubations can be treated as if acquired by an in-situ flow through system.

For the interpretation of the measured concentration differences, the multifaceted influence of metabolic processes on chemical fluxes was accounted for. The model shows a good fit of the observations around the fitted curve, indicating that the interactions between processes and their effects on chemical fluxes were considered correctly.

Nonetheless, due the limited number of incubations that were carried out for this study, we interpret results with caution. Additionally, incubations were only deployed on the reef flat of one degraded reef site, future application of this or similar

incubation methods should consider multiple sites. Lastly, methods would be further improved by continuous monitoring of the exchange rate, rather than assuming it to be constant throughout the incubation.

**Conclusions**

Results acquired on this shallow Curaçaoan reef provide insight into the impact of habitat degradation and benthic composition shifts on reef functions. Currently prevalent corals, although more resilient, calcify at a slower pace than previously abundant species (such as *Acropora* spp.) and cannot balance out heterotrophic processes from other functional groups. Coral presence does however contribute to counteracting dissolution processes at night, therefore acting as buffers, albeit marginal, to reef deconstruction. In the context of ongoing global change, the environmental resilience of generalist species could be a

determining factor of ecosystem stability (Clavel et al. 2011). For instance, on some reef terrace of the fringing reefs of Curaçao and Bonaire (southern Caribbean), certain stretches appear to harbour a considerable cover of steadily growing little boulder-constructing tolerant corals (including mainly (*Pseudo*)*Diploria* spp., *Porites astreoides* and *Siderastrea sidereal*) (de Bakker et al. 2019). These are often found near areas which have locally suffered chronic stress from terrestrial sources (i.e., inflow, intense coastal development, factory outflow) but often limited to areas providing hard substrate and relatively little sand. Data

on the processes underlying such developments however is virtually absent, but this may indicate that even the most severely degraded reefs could slowly regain essential functions when a critical adaptive capacity is reached.

While these sites may provide a spark of hope with regards to recovery potential, most of the reefs in the Caribbean presently reside in ecological states that closely resemble the reef site studied here (or are expected to do so in the near future). Ultimately, Caribbean reefs will benefit most notably from adequate mitigation strategies to give these systems a chance to adapt and

restore key functions in the face of exacerbating environmental conditions.

**Data and code availability** Data and R code will be made available on request.

**Author's contributions** AW, DdB, SvH and LdN conceived the ideas and designed methodology; AW, DdB and TdC

collected the data; AW and KS analysed the data; AW and LdN led the writing of the manuscript in consultation with DdB, SvH, FvD, KS and GR. All authors contributed critically to the drafts and gave final approval for publication.

**Competing Interests** The authors declare that they have no conflict of interest.

**Acknowledgements** The authors are particularly grateful to the Carmabi research station as a whole and especially Mark Vermeij for his support and facilitating the fieldwork on Curaçao. We would also like to thank Jasper de Goeij for lending us equipment. We extent our sincere gratitude to Karel Baker and Sharyn Ossebaar for analysing nutrient and DIC samples.

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

**Figure 1: The tent incubation setup used. (a) Photograph depicting the tent incubation during the experiments. (b) Schematic cross section of the employed setup for enclosing a small patch of reef. A battery powered mixing propeller for maintaining water**
**circulation, and analysers for salinity (S), temperature (T), oxygen (O$_2$), and light (PAR) are located inside the tent. Outside the enclosure another S/T and PAR analyser were placed, as well as the battery for the pump. Sampling of exterior and interior water (though sampling tube) was performed by divers using large volume syringes. Zippers allow for opening of tent windows for re-equilibrating the interior to the exterior conditions between incubations.**

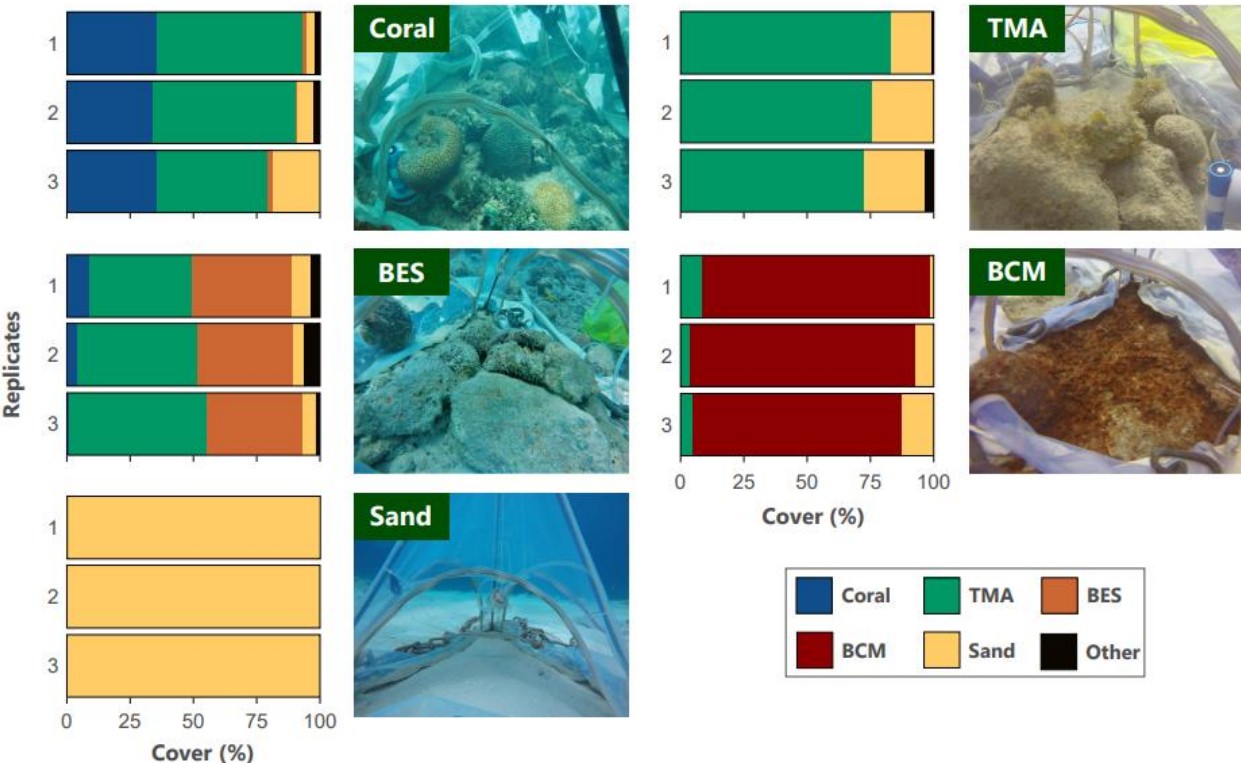

Figure 2: Benthic cover of the 15 incubated reef communities dominated either by coral, turf and macroalgae (TMA), bioeroding sponges (BES), cyanobacterial mats (BCM) or sand with exemplary photograph.

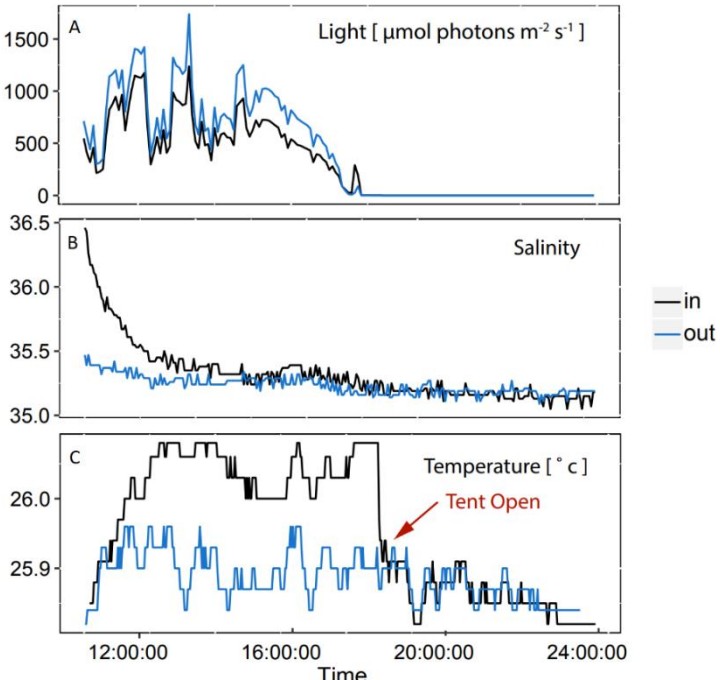

**Figure 3: Exterior and interior measurements for three representative incubations performed throughout this study. (a) Difference in light between the inside of the tent and the ambient environment during an incubation. (b) Injection of salt within the tent at the start of the incubation and its gradual return to ambient salinity. (c) Temperature within the tent compared to exterior conditions, when the tent is opened at the end of the incubation, temperature immediately returns to ambient conditions.**

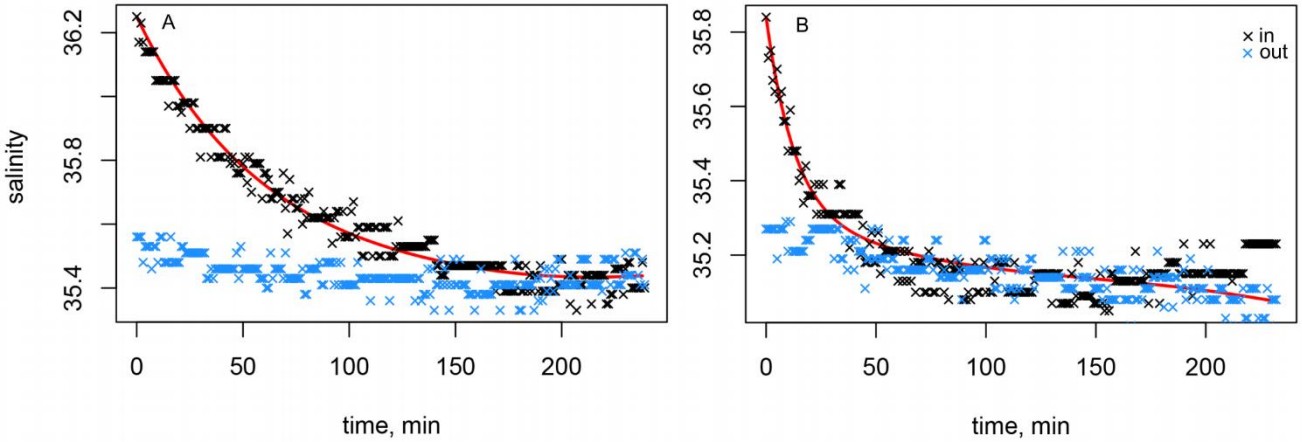

**Figure 4: (a, b) Best fit explaining in-tent salinity at any given time (equation 1, red line). In-tent salinity measurements are in black and ambient salinity measurements are in blue. (a) illustrates an incubation that leaked relatively slowly with $K = 0.0192$ ($0.0192 * 108 = 2.1$ litres per minute), while (b) depicts a more rapidly leaking tent with $K = 0.044$ ($0.044 \times 108 = 4.8$ litres per minute).**

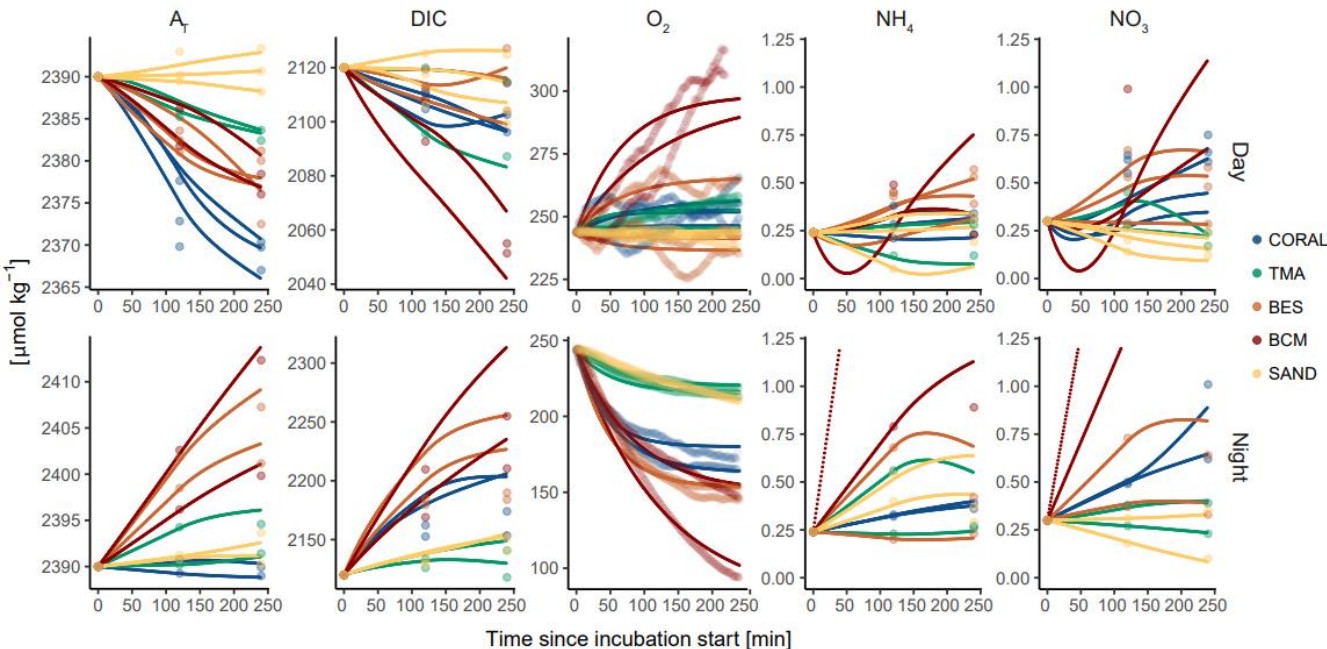

Figure 5: Illustrative results of the model (coloured lines) employed to infer the process rates from measured data. The top graphs depict the model output of incubations performed during the day while the bottom ones show the output for night-time incubations. Points represent measured values of the state variables inside and outside the tent. The measured data was centred for graphic visualisation purposes (non-centred data for all incubations can be found in supplementary Table S2). The blue, green, orange, red and yellow colours represent communities dominated by coral, turf and macroalgae, bioeroding sponges, benthic cyanobacterial mats and sand respectively. Note that $NH_4$ and $NO_3$ measurements in BCM incubations at night-time were much higher than the rest. The y-axis of the graphs depicting $NH_4$ and $NO_3$ results were therefore truncated in order to better visualise model output of all other incubations (model output for the off chart BCM incubation can be found in Figure S1).

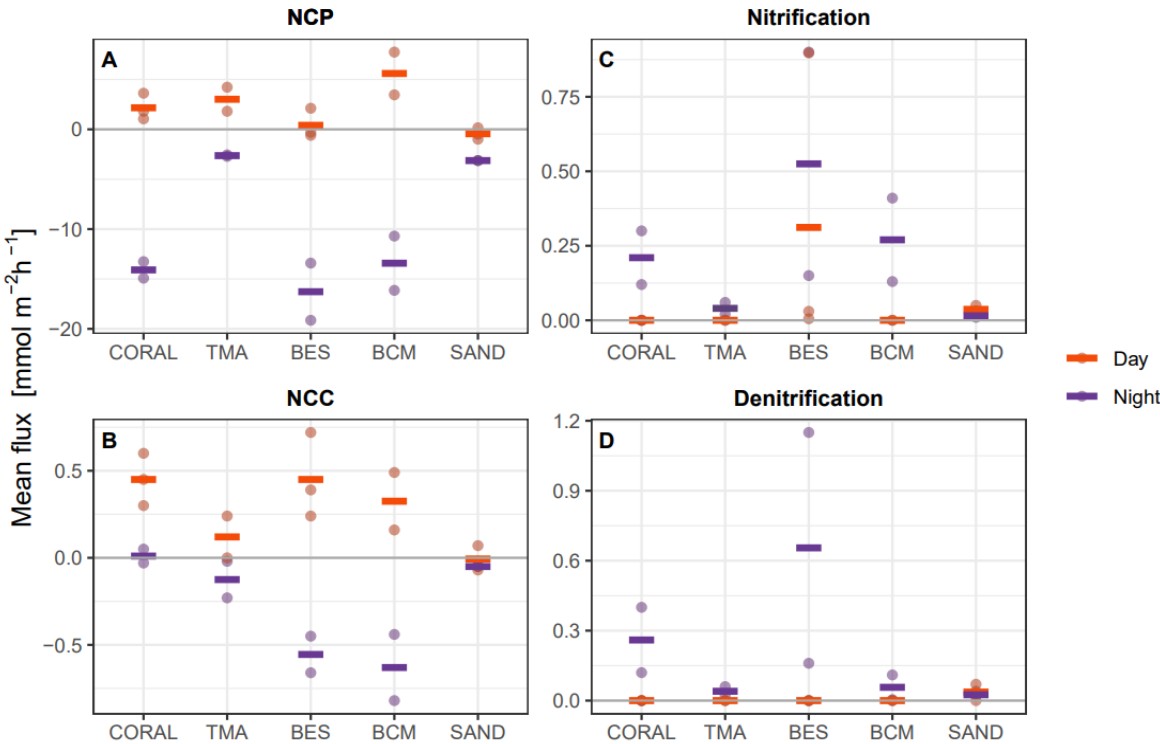

**Figure 6: Process rates for all tent replicates (points) and respective average (line) estimated from observed concentration changes and model output in the tent enclosure on each substrate type. Processes occurring during the day are depicted in red, and processes at night are represented in purple.**

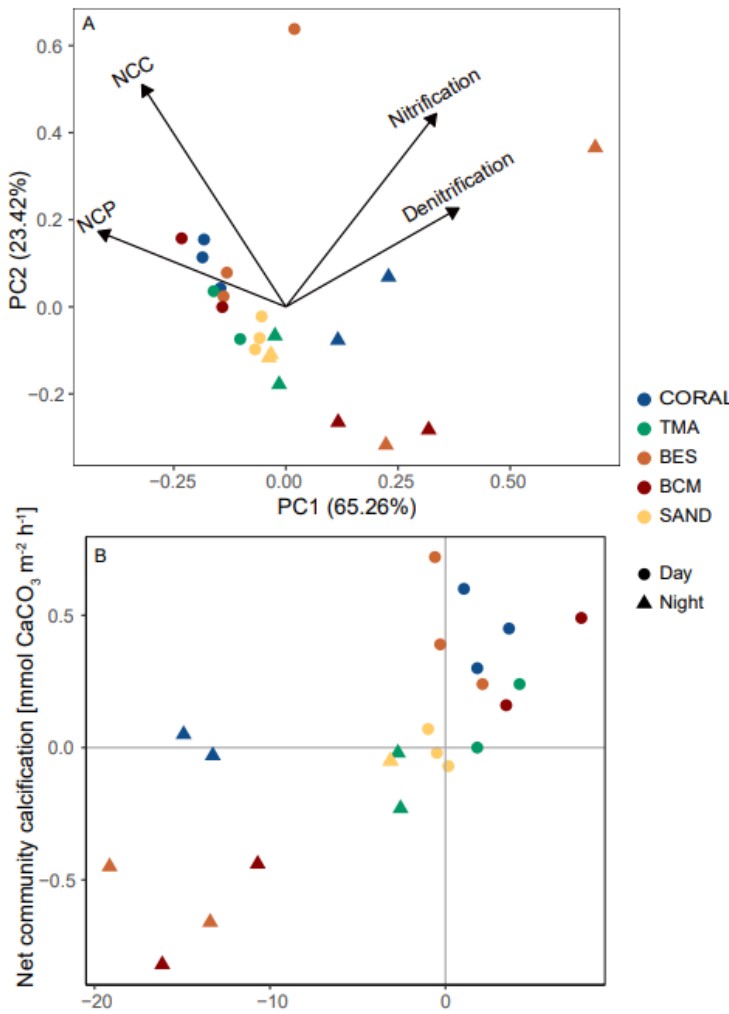

**Figure 7: A) Principal component analysis (PCA) diagram displaying the spatial variation of all incubations along the first two principal components. Processes are plotted as vectors and dots represent day-time incubations while triangles depict night-time incubations. Colour refers to the type of substrate incubated.**

**B) NCP vs NCC rates for every incubation. Each dot/triangle represents an individual incubation and colour indicates community composition.**