# Peer review of "Quantifying functional consequences of habitat degradation on a Caribbean coral reef"

_Biogeosciences, 2021_

## Referee Comment (RC1)

Webb et al Biogeosciences

This is a very interesting paper that addresses biogeochemistry among five different substrate types in the Caribbean using closed in situ tent incubations. There is a wealth of data here that would be great to see published in Biogeosciences. However, I have several key concerns in the methods that should be addressed before this work is considered, mostly relating to the low units of replication, and consequent analyses, narrative and overall extrapolation of reef functioning thereafter.

It seems that the inverse model outputs employed distract from more targeted analyses and presentation of, what could be, quite compelling results on biogeochemical fluxes in the tents among substrate types. This modelling approach may have been adopted owing to the low sample size in the study; n=3 daytime and n=2 nighttime per substrate type. If so, the authors should express this clearly. While I cannot speak for the models themselves, as this is far from my expertise, I am concerned that the models were established based on just 2-3 replicates per treatment. In fact, calculations of 95% CIs (e.g. Fig 6) were also inferred from model outputs owing to low sample size. It is likely that the low level of replication within each substrate type (n=2-3) skews the overall result towards "no significant differences between any communities" (Ln 243), when this is not actually the case. This appears to be a key take-home message (e.g. abstract Ln 28; "no significant difference between processes on any assemblage") that may not be true, but there was not enough power to show otherwise. This forms a narrative and conclusion that may dangerously misrepresent the data, which poses significant risk when extrapolating findings to ecosystem functioning and functional redundancy.

The inverse modelling seems to overlook specific results among treatments. Table S2 presents model output data, but this does not compare treatments/factors (day vs. night, substrate types). Such comparative analyses are found in Table S3, which is the PERMANOVA result that shows no significant differences, except for day vs. night. It is unclear what data (factors or response variables) were even used for the PERMANOVA, which I feel may not be the correct approach to analysing these data.

I would argue that if more targeted analyses (e.g. linear models, ANOVA) were conducted, say for NCP or NCC between day-night and substrate types, more informative results would emerge. For example, in Fig 6, NCP is much lower at night in coral, BES and BCM than in sand and TMA. Conversely, NCC is greater in the day for coral, BES and BCM than in sand, and lowest at night for BES and BCM. I am just eyeballing data here, but this does not look like "no significant differences between any communities" (Ln 243). I am not sold by the PERMANOVA, as it seems there should be some detectable differences between substrate types, and that greater detail and resolution could uncover this.

Also, it seems that the dominant substrate type was used to categorise factors, but this may impact (and limit) the results due to low sample size. One alternative to this issue could be to analyse all tents using continuous data for substrate type. E.g. could data be analysed at the level of "percent cover of sand", "percent cover of coral", "percent cover of cyanobacteria", and so on… rather than treating them as fixed categorical factors? This would increase sample size, and possibly tease out interesting results e.g. thresholds of cover for positive or negative results regarding ecosystem processes and functions. Otherwise, more data / replicates may be needed. I fear that in using the current approach, conclusions are made on ecosystem functioning and redundancy that extend beyond the scope of the data.

**Abstract**
Ln 23-33: I suggest changing to past tense here, e.g. "Estimated processes **were** low" and "No real gain in primary habitat **was** recorded"…. And so on.

Ln 28: Suggest removing reference to the analysis here "A multivariate pairwise analysis revealed that there is no significant difference…" to be more succinct, e.g. "We found no significant difference…"

**Introduction**
Ln 36: "habita**nt**" should be "habitat"

The authors should consider other relevant literature on coral reef ecosystem functioning and functional redundancy, e.g.
- Range of work by Bellwood, e.g.:
    o Bellwood, D. R., Hoey, A. S., & Choat, J. H. (2003). Limited functional redundancy in high diversity systems: resilience and ecosystem function on coral reefs. *Ecology letters*, *6*(4), 281-285
    o Bellwood, D. R., Streit, R. P., Brandl, S. J., & Tebbett, S. B. (2019). The meaning of the term 'function' in ecology: a coral reef perspective. *Functional Ecology*, *33*(6), 948-961
- Wolfe, K., Anthony, K., Babcock, R. C., Bay, L., Bourne, D. G., Burrows, D., ... & Mumby, P. J. (2020). Priority species to support the functional integrity of coral reefs. *Oceanography and Marine Biology*.

Ln 70: Additional work could be considered including in situ and lab experiments, e.g.
- Albright, R., Caldeira, L., Hosfelt, J., Kwiatkowski, L., Maclaren, J. K., Mason, B. M., ... & Caldeira, K. (2016). Reversal of ocean acidification enhances net coral reef calcification. *Nature*, *531*(7594), 362-365
- Albright, R., Takeshita, Y., Koweek, D. A., Ninokawa, A., Wolfe, K., Rivlin, T., ... & Caldeira, K. (2018). Carbon dioxide addition to coral reef waters suppresses net community calcification. *Nature*, *555*(7697), 516-519
- Dove, S. G., Kline, D. I., Pantos, O., Angly, F. E., Tyson, G. W., & Hoegh-Guldberg, O. (2013). Future reef decalcification under a business-as-usual $CO_2$ emission scenario. *Proceedings of the National Academy of Sciences*, *110*(38), 15342-15347
- Dove, S. G., Brown, K. T., Van Den Heuvel, A., Chai, A., & Hoegh-Guldberg, O. (2020). Ocean warming and acidification uncouple calcification from calcifier biomass which accelerates coral reef decline. *Communications Earth & Environment*, *1*(1), 1-9
- Brown, K. T., Bender-Champ, D., Achlatis, M., van Der Zande, R. M., Kubicek, A., Martin, S. B., ... & Hoegh-Guldberg, O. (2021). Habitat-specific biogenic production and erosion influences net framework and sediment coral reef carbonate budgets. *Limnology and Oceanography*, *66*(2), 349-365

**Methods**
Ln 92, 93, 95, 144, 150, 184, etc… Suggest making methods section past tense e.g. change "is" to "was", "are" to "were", etc.

Ln 102: More information should be provided on the number of replicate tents used per substrate type;
- There were five substrate treatments, each with three replicates (?) (Fig 2). Were all of these deployed at the same time or was a single custom tent reused for all?
- How long were tents left over the substrate before beginning the experiments / incubations?
- Was the substrate left to stabilise in cases where substrate was moved into the tent to artificially construct the benthic community?
- Daytime incubations were done in triplicates (Ln 102) and night incubations in duplicates (Ln 102), but is this n=3 tents per substrate incubated 3 (day; n=9) and 2 (night; n=6) times **or** just one tent done n=3 (day) and n=2 (night) times?

   o If the former, was tent number incorporated as a factor to account for psuedoreplication of tent/substrate type? And were there differences detected in seawater parameters across incubations (i.e. repeated measures?)

   o If the latter, units of replication per treatment and timepoint are quite low.

Ln 103: Incubations went for 4 hrs each, but did they all start around the same time day and night? It seems in Fig 5 that O2 does not follow the trend line selected, but instead has large fluxes across the 4 hrs. Does this reflect differences in incubation time, perhaps started later in the morning or afternoon than others..? Perhaps add a sentence like "all daytime incubations were started at X am, and all nighttime incubations were started at X pm".

Ln 116: Species name must be italicised

Ln 134: Were filters changed between samples and incubation time points? If so, how? If not, could sediment and particles trapped in the filters from T0 have impacted samples at T2 and T4?

Ln 203-207: Be clear if benthic cover (dominance) was used as a fixed categorical factor in analyses. If it was, how may the difference in cover for algal dominance (72-83%) or cyanobacteria dominance (83-91%) have influenced results from these respective tents? I believe that a 10% difference in algal or cyanobacteria cover could be quite influential. This seems like an important consideration given that there were possibly just n=3 replicates per substrate type. (See major comment above).

Ln 205: Was PERMANOVA conducted on raw or model data? This is important to state. I am not sure how robust this analysis is to such low sample size n=2 within raw data, but also unsure whether such analyses should be conducted on modelled data. Further, what biochemical processes (variables) were analysed using PERMANOVA? Table S3 shows just one set of output data, but how does this translate to NCC, NCP, etc? What exactly was tested here?

**Results**
The results would benefit from a few subheadings to form structure. E.g. #1 general temperature, salinity, light in the tents – baseline conditions / tent effects, #2 incubations with differences in AT, CT, pH, N, etc. among tents and substrate, and #3 NEC and NCP among tents and substrates.

Ln 215-218: How did these leakages impact incubations? Given the exchange rates were greater in these two tents, is it possible that their leaking confounded the results from these tents? How was this accounted for?

Ln 227, 228, 231, etc: Again, I would stick to past tense in results section, e.g. "NCP **showed** a clear diurnal pattern"

Figure 5: This figure looks like a copy-paste model output. Minimum, panels should be better labelled. However, a more informative figure could be produced that summarises the model outputs for the five substrates, day and night.

Figure 6: Panels should be labelled with A, B, C and D. Also, it now seems that Table 1 is redundant given this information is provided here in Figure 6. It is much easier to view as a figure, so I suggest deleting Table 1.

On this note, it would be nice to see other seawater chemistry data (AT, CT, pH, etc) presented like this in a separate figure or table. I feel the results are short and overlook

baseline measurements of seawater chemistry among tents and substrate types. How did AT, CT, pH vary among substrates and across incubations?

Ln 242: "per" should be capitalised in PERMANOVA

**Discussion**
Ln 253: This is a nice contrast, but make it clear by stating the ranges of NCC you found, as well as that commonly found in the literature.
Ln 254: What does "no real gain in primary habitat" mean? Do you mean reef accretion / coral growth? If so, use this very carefully, as NCC and accretion are not always coupled. Low (or high) rates of NCC do not always correspond to low (or high) accretion rates. Did you measure primary habitat gain somehow, or is this assumed from NCC rates?
Ln 255: As above, saying "accumulation of biomass through photosynthesis is low" may not be true.
Also, use past tense throughout: "was" not "is", "were" not "are"

Ln 260: I am not convinced by this statement. Fig 6 shows differences among substrates, which more explicit analyses may reveal. Functional homogenisation is quite a loaded term to use from n=2-3 replicates.

Ln 267, 280, 298, etc: Consider a new paper that addresses calcification and dissolution on dead and live coral surfaces:
- Romanó de Orte, M., Koweek, D. A., Cyronak, T., Takeshita, Y., Griffin, A., Wolfe, K., ... & Caldeira, K. (2021). Unexpected role of communities colonizing dead coral substrate in the calcification of coral reefs. *Limnology and Oceanography*

Ln 269: In reference to my comment above, how was "accretion rate" calculated?

Ln 274: "an" should be "and"
Ln 275: "Koweet" should be "Koweek"

Ln 290-293: This text is useful but has no clear point as currently expressed. I also see its relevance at Ln 365.

Ln 341: As above, this key message may not be correct given the low sample size and unclear PERMANOVA analysis. The narrative and analyses must be readdressed.

Ln 345: This level of 34-36% coral cover is very high for the Caribbean. Why was this done? Were corals intentionally moved into tents to create this high coral cover? If so, were they left to stabilise for several days after relocation? Were any metrics of coral condition measured before (and/or after) incubations? Is it possible that the corals were stressed and under-performing?

---

## Author Comment (AC4)

[Figure]

**Figure 5: Illustrative results of the model (coloured lines) employed to infer the process rates from measured data. The top graphs depict the model output of incubations performed during the day while the bottom ones show the output for night-time incubations. Points represent measured values of the state variables inside and outside the tent. The measured data was centred for graphic visualisation purposes (non-centred data for all incubations can be found in supplementary Table S2). The blue, green, orange, red and yellow colours represent communities dominated by coral, turf and macroalgae, bioeroding sponges, benthic cyanobacterial mats and sand respectively. Note that $NH_4$ and $NO_3$ measurements in BCM incubations at night-time were much higher than the rest. The y-axis of the graphs depicting $NH_4$ and $NO_3$ results were therefore truncated in order to better visualise model output of all other incubations (model output for the off chart BCM incubation can be found in Figure S1).**

[Figure]

**Figure 6: Process rates for all tent replicates (points) and respective average (line) estimated from observed concentration changes and model output in the tent enclosure on each substrate type. Processes occurring during the day are depicted in red, and processes at night are represented in purple.**

[Figure]

**Figure 7: A)** Principal component analysis (PCA) diagram displaying the spatial variation of all incubations along the first two principal components. Processes are plotted as vectors and dots represent day-time incubations while triangles depict night-time incubations. Colour refers to the type of substrate incubated.
**B)** NCP vs NCC rates for every incubation. Each dot/triangle represents an individual incubation and colour indicates

---

## Author Response (AR1)

Dear dr. Cyronak

       On behalf of all authors, we thank you for your constructive comment on our manuscript and are happy to present an updated version, also taking into account the comments by the two reviewers. In your comment, you suggest to either remove the modelling part or justify its position in the overall study. Since removing it will compromise the essence of our research, we suggest clarifying, here and in the manuscript, the (added) value of the modelling approach in relation to the results.

We are using this model to derive process rates such as nitrification and calcification rather than calculating fluxes directly from measured chemistry data. As such, the model inferred rates, as presented in the manuscript, are not the product of one measured variable but derived from the interaction of multiple components. We have stressed the advantage of this approach in the manuscript because it constrains the actual fluxes much better. Specifically, it circumvents the potentially large errors in calculated fluxes when relying on only one measured parameter.

For instance, nitrification rates cannot simply be quantified by the measured increase or decrease in $NH_4$, because a multitude of processes are simultaneously impacting $NH_4$ fluxes, as represented by the equation below:

$$\frac{dNH_4}{dt} = mineralisation \times NCratio - PP \times NCratio \times Pnh4 - nitrification + K\ (NH_4out - NH_4)$$

In addition, nitrification itself impacts the fluxes of $O_2$ and $NO_3$:

$$\frac{dO_2}{dt} = -mineralisation \times OC_{ratio} \times (1 - pDeni)\ + PP \times OC_{ratio} - 2 \times nitrification + K\ (O_2meanout - O_2)$$

$$\frac{dNO_3}{dt} = -(mineralisation \times pDeni \times 0.8) - PP \times NC_{ratio} \times (1 - Pnh4) + nitrification + K\ (NO_3out - NO_3)$$

Our methodology enables us to account for the simultaneous impacts that the various processes have on the measured variables and allows us to relate the change in their concentrations to the responsible metabolic processes.

We have added a clarification to why we use this method in the abstract of the revised manuscript l.22:

"To account for the simultaneous influence that distinct biogeochemical processes have on measured fluxes, the rates are then derived by solving a model consisting of differential equations describing the contribution of each process to the measured chemical fluxes."

We sincerely hope that this clarification justifies the model's contribution to our work.

We would also like to thank the two anonymous reviewers for the overall feedback and the helpful comments which have improved the quality of this manuscript. Please find below our point-by-point answers to the reviewer's comments, with purple text indicating text revised or added to the newest version of the manuscript. We enclose a "track changes" version of the manuscript with revisions highlighted, as well as a "clean" version. For your convenience, line numbers referred to in our responses correspond to lines in the "track changes" version of the revised manuscript.

We look forward to your response.

Very best regards,

Alice Webb and co-authors

Revision **of submission titled:**

'Functional consequences of Caribbean coral reef habitat degradation'

Authors:

Alice E. Webb, Didier M. de Bakker, Karline Soetaert, Tamara da Costa, Steven M. A. C. van Heuven, Fleur C. van Duyl, Gert-Jan Reichart, Lennart J. de Nooijer

*Corresponding author:
Alice Webb, webbea4@gmail.com

**Comments from reviewers:**

| Referee #1 (Comments to the Author): |
| --- |

This is a very interesting paper that addresses biogeochemistry among five different substrate types in the Caribbean using closed in situ tent incubations. There is a wealth of data here that would be great to see published in Biogeosciences. However, I have several key concerns in the methods that should be addressed before this work is considered, mostly relating to the low units of replication, and consequent analyses, narrative and overall extrapolation of reef functioning thereafter.

It seems that the inverse model outputs employed distract from more targeted analyses and presentation of, what could be, quite compelling results on biogeochemical fluxes in the tents among substrate types. This modelling approach may have been adopted owing to the low sample size in the study; n=3 daytime and n=2 nighttime per substrate type. If so, the authors should express this clearly. While I cannot speak for the models themselves, as this is far from my expertise, I am concerned that the models were established based on just 2-3 replicates per treatment. In fact, calculations of 95% CIs (e.g. Fig 6) were also inferred from model outputs owing to low sample size. It is likely that the low level of replication within each substrate type (n=2-3) skews the overall result towards "no significant differences between any communities" (Ln 243), when this is not actually the case. This appears to be a key take-home message (e.g. abstract Ln 28; "no significant difference between processes on any assemblage") that may not be true, but there was not enough power to show otherwise. This forms a narrative and conclusion that may dangerously misrepresent the data, which poses significant risk when extrapolating findings to ecosystem functioning and functional redundancy.

✓ We thank the reviewer for this relevant remark. The modelling approach used in this study to estimate biogeochemical processes was first and foremost to relate the measured change in concentrations of variables ($A_T$, DIC, etc.) to the responsible metabolic processes. The reviewer is correct in pointing out that the limited number of replicates may be insufficient to permit the use of a PERMNOVA to analyse our data. Although we group the incubations in five different categories based on the dominant group, we

do acknowledge that these communities do harbour differences in terms of composition and therefore refrain from performing analysis using the community composition as a categorical factor.

In line with the reviewer's key concerns:

- we removed the claim of functional homogenisation
- altered the title to put a bit of emphasise on the methodology aspect of the paper, it now reads:

'*Quantifying functional consequences of habitat degradation on a Caribbean coral reef*'

- We removed the PERMANOVA analysis (see below) and replaced it by PCA was conducted on a centred multivariate data set consisting of the four main biogeochemical processes (i.e., NCC, NCP, nitrification and denitrification).
- We rephrased some parts of the introduction and discussion and included a more cautious interpretation of the results.

- Although for visual ease, we do group the incubations in 5 different categories (with various colours), we refrained from performing analysis using composition as a categorical factor, rather treating each incubation as an individual.
- The reviewer is correct in pointing out that calculating 95% CI on only 2 replicates is not relevant. Instead, we plot figure 6 as a scatter plot depicting all estimated parameters with their mean.

The inverse modelling seems to overlook specific results among treatments. Table S2 presents model output data, but this does not compare treatments/factors (day vs. night, substrate types). Such comparative analyses are found in Table S3, which is the PERMANOVA result that shows no significant differences, except for day vs. night. It is unclear what data (factors or response variables) were even used for the PERMANOVA, which I feel may not be the correct approach to analysing these data.

✓ The PERMANOVA analysis has been removed.

I would argue that if more targeted analyses (e.g. linear models, ANOVA) were conducted, say for NCP or NCC between day-night and substrate types, more informative results would emerge. For example, in Fig 6, NCP is much lower at night in coral, BES and BCM than in sand and TMA. Conversely, NCC is greater in the day for coral, BES and BCM than in sand, and lowest at night for BES and BCM. I am just eyeballing data here, but this does not look like "no significant differences between any communities" (Ln 243). I am not sold by the PERMANOVA, as it seems there should be some detectable differences between substrate types, and that greater detail and resolution could uncover this.

✓ The PERMANOVA was removed. Due to limited numbers of incubations, we also didn't perform an ANOVA using substrates as a categorical factor.

Instead, we performed a PCA (now figure 7) based on the four main biogeochemical processes estimated for each incubation. We added text to the methods l. 248:

*Principal component analysis (PCA) was used to identify grouping among the 23 tent incubations (day n = 13, night: n = 10) in relation to their biogeochemical signature (i.e., NCC, NCP, nitrification and*

*denitrification). PCA was conducted on a centred multivariate data set consisting of the four main biogeochemical processes (i.e., NCC, NCP, nitrification and denitrification).*

And in the results l. 312:

*The PCA based on the four main biogeochemical processes revealed two main different groups between night incubations and day incubations (Figure 7A). Sand incubations were the exception as night and day incubations grouped relatively close to each other. The first two principal component axes (PC1 and PC2) explained 88.68% of the total variability within the data. PC1 described a gradient in NCP and NCC from high (negative PCA scores) to low (positive PCA scores) and an opposite pattern for nitrification and denitrification. PC2 further explains the variability in NCC and nitrification, and to a lesser extent NCP and denitrification. One of the communities dominated by bioeroding sponges (rep.1) is separate from other communities both during the day and during the night due to high rates of nitrification and denitrification compared to other communities.*

Also, it seems that the dominant substrate type was used to categorise factors, but this may impact (and limit) the results due to low sample size. One alternative to this issue could be to analyse all tents using continuous data for substrate type. E.g. could data be analysed at the level of "percent cover of sand", "percent cover of coral", "percent cover of cyanobacteria", and so on… rather than treating them as fixed categorical factors? This would increase sample size, and possibly tease out interesting results e.g. thresholds of cover for positive or negative results regarding ecosystem processes and functions. Otherwise, more data / replicates may be needed. I fear that in using the current approach, conclusions are made on ecosystem functioning and redundancy that extend beyond the scope of the data.

✓ Although we considered this proposition very seriously, and tried out potential analysis accordingly, the present experiment protocol was not built to answer this question and therefore analysing data at the level of percentage of sand/bcm/coral… is problematic as the percentage of cover of different groups in the incubations is not continuous. For instance, coral is only present in coral and bioeroding sponge dominated communities, and cyanobacteria are only present in BCM dominated tents. Getting thresholds of cover for positive or negative results regarding ecosystem processes is challenging due to the large gaps between percentage cover of various groups (e.g., for sand, the highest cover is 100 % followed by 24 %). Hence, we refrained from such an analysis.

**Abstract**

Abstract Ln 23-33: I suggest changing to past tense here, e.g. "Estimated processes were low" and "No real gain in primary habitat was recorded"…. And so on.

✓ We have made changes accordingly.

Ln 28: Suggest removing reference to the analysis here "A multivariate pairwise analysis revealed that there is no significant difference…" to be more succinct, e.g. "We found no significant difference…

✓ We removed this sentence because we removed the PERMANOVA. Instead, we write at l. 33:

*Results suggest similar directions and magnitudes of key biogeochemical processes of distinct communities on this shallow Curaçaon reef.*

**Introduction**

Ln 36: "habitant" should be "habitat"

✓ We have made changes accordingly.

The authors should consider other relevant literature on coral reef ecosystem functioning and functional redundancy, e.g. - Range of work by Bellwood,

e.g.: o Bellwood, D. R., Hoey, A. S., & Choat, J. H. (2003). Limited functional redundancy in high diversity systems: resilience and ecosystem function on coral reefs. Ecology letters, 6(4), 281-285

Bellwood, D. R., Streit, R. P., Brandl, S. J., & Tebbett, S. B. (2019). The meaning of the term 'function' in ecology: a coral reef perspective. Functional Ecology, 33(6), 948-961 - Wolfe, K., Anthony, K., Babcock, R.

C., Bay, L., Bourne, D. G., Burrows, D., ... & Mumby, P. J. (2020). Priority species to support the functional integrity of coral reefs. Oceanography and Marine Biology.

✓ We thank the reviewer for these relevant publications, we added them to the introduction l. 65, 76 and 63 respectively.

Ln 70: Additional work could be considered including in situ and lab experiments,

e.g. - Albright, R., Caldeira, L., Hosfelt, J., Kwiatkowski, L., Maclaren, J. K., Mason, B. M., ... & Caldeira, K. (2016). Reversal of ocean acidification enhances net coral reef calcification. Nature, 531(7594), 362-365

- Albright, R., Takeshita, Y., Koweek, D. A., Ninokawa, A., Wolfe, K., Rivlin, T., ... & Caldeira, K. (2018). Carbon dioxide addition to coral reef waters suppresses net community calcification. Nature, 555(7697), 516-519

- Dove, S. G., Kline, D. I., Pantos, O., Angly, F. E., Tyson, G. W., & Hoegh-Guldberg, O. (2013). Future reef decalcification under a business-as-usual CO2 emission scenario. Proceedings of the National Academy of Sciences, 110(38), 15342-15347

- Dove, S. G., Brown, K. T., Van Den Heuvel, A., Chai, A., & Hoegh-Guldberg, O. (2020). Ocean warming and acidification uncouple calcification from calcifier biomass which accelerates coral reef decline. Communications Earth & Environment, 1(1), 1-9

- Brown, K. T., Bender-Champ, D., Achlatis, M., van Der Zande, R. M., Kubicek, A., Martin, S. B., ... & Hoegh-Guldberg, O. (2021). Habitat-specific biogenic production and erosion influences net framework and sediment coral reef carbonate budgets. Limnology and Oceanography, 66(2), 349-365

✓ We thank the reviewer for these publications, we added these papers to the introduction l. 79,79, and 86,86 respectively.

**Methods**

Ln 92, 93, 95, 144, 150, 184, etc… Suggest making methods section past tense e.g. change "is" to "was", "are" to "were", etc

✓We have made changes accordingly

Ln 102: More information should be provided on the number of replicate tents used per substrate type;

- There were five substrate treatments, each with three replicates (?) (Fig 2).

✓Yes, we have rephrased the sentence l. 121. It now read:

Incubated communities included five different types of substrate dominated either by turf and macroalgae (n=3), sand (n=3), bioeroding sponges (n=3), benthic cyanobacteria mats or coral (Fig. 2), equalling a total of 15 studied communities. Each community was incubated during the day (n=15) and due to practical reasons, only 2 of each type were incubated during the night (n=10) (i.e., for each type of community, three daytime and 2 night-time incubations were carried out).

Were all of these deployed at the same time or was a single custom tent reused for all?

✓We used two custom tents which we interchanged so that they both spent some time out of the water to avoid fouling. Only one incubation was performed at a time due to limited Oxygen, CTD and pump equipment.

How long were tents left over the substrate before beginning the experiments / incubations?

✓Prior to each incubation, the tent was place with flaps open over the substrate and lefts for a minimum of 3 hours before the incubation was started to permit the community to get to acclimatise and sand to settle. We have added a paragraph with this information (also for the question above) l.126:

*The incubations were carried out one at a time, over the study period and lasted four hours each. Prior to each incubation, the tent was place with flaps open over the substrate and lefts for a minimum of 3 hours before the incubation was started. When day incubations were terminated, the tent was left in place with flaps open until the night incubation was carried out on the same substrate. All daytime incubations were started at 10:00 and all night-time incubations were started at 18:30.*

Was the substrate left to stabilize in cases where substrate was moved into the tent to artificially construct the benthic community?

✓*In these cases, the community was left to stabilise two or three days before starting incubations.* This information is now added l.135.

Daytime incubations were done in triplicates (Ln 102) and night incubations in duplicates (Ln 102), but is this n=3 tents per substrate incubated 3 (day; n=9) and 2 (night; n=6) times or just one tent done n=3 (day) and n=2 (night) times?

o If the former, was tent number incorporated as a factor to account for pseudo-replication of tent/substrate type? And were there differences detected in seawater parameters across incubations (i.e. repeated measures?)

o If the latter, units of replication per treatment and timepoint are quite low.

✓To make this clearer, we have added the paragraph to section 2.2 (lines 120 and on):

*Incubated communities included five different types of substrate dominated either by turf and macroalgae (n=3), sand (n=3), bioeroding sponges (n=3), benthic cyanobacteria mats or coral (Fig. 2), equalling a total of 15 studied communities. Each community was incubated during the day (n=15) and due to practical reasons, only 2 of each type were incubated during the night (n=10) (i.e., for each type of community, three daytime and 2 night-time incubations were carried out).*

Ln 103: Incubations went for 4 hrs each, but did they all start around the same time day and night? It seems in Fig 5 that O2 does not follow the trend line selected, but instead has large fluxes across the 4 hrs. Does this reflect differences in incubation time, perhaps started later in the morning or afternoon than others..? Perhaps add a sentence like "all daytime incubations were started at X am, and all nighttime incubations were started at X pm".

✓Yes, day incubation all started at 10:00 and night incubation at 18:30. We have added this to l. 129.

*All daytime incubations were started at 10:00 and all night-time incubations were started at 18:30.*

Ln 116: Species name must be italicized

✓We italicised the species *Oscillatoria bonnemaisonii* now l. 143.

Ln 134: Were filters changed between samples and incubation time points? If so, how? If not, could sediment and particles trapped in the filters from T0 have impacted samples at T2 and T4?

✓Filters were changed regularly but not during incubations. When filters were changed, they were inspected and did not show over saturation of sediment. We visually inspected every single alkalinity samples before processing them on the optical titrator. The sand found at this site consists mostly of carbonate. Even with small grains of sediment present in the samples, they will dissolve after addition of acid and result in pronounced peaks in alkalinity. Since this did not occur, we are positive there was no sediment in the samples. We have added the sentence below to explain these filters were changes regularly l. 162.

*The tube end located inside the tent was equipped with a Whatman ® filter (G/F 0.47 µm) which was replaced daily.*

Ln 203-207: Be clear if benthic cover (dominance) was used as a fixed categorical factor in analyses. If it was, how may the difference in cover for algal dominance (72-83%) or cyanobacteria dominance (83-91%) have influenced results from these respective tents? I believe that a 10% difference in algal or cyanobacteria cover could be quite influential. This seems like an important consideration given that there were possibly just n=3 replicates per substrate type. (See major comment above).

✓As the reviewer recommend, we have changed our statistical analysis. See answer to major comment above.

Ln 205: Was PERMANOVA conducted on raw or model data? This is important to state. I am not sure how robust this analysis is to such low sample size n=2 within raw data, but also unsure whether such analyses should be conducted on modelled data. Further, what biochemical processes (variables) were analysed using PERMANOVA? Table S3 shows just one set of output data, but how does this translate to NCC, NCP, etc? What exactly was tested here?

✓We have removed the PERMANOVA analysis, see above.

**Results**

The results would benefit from a few subheadings to form structure. E.g. #1 general temperature, salinity, light in the tents – baseline conditions / tent effects, #2 incubations with differences in AT, CT, pH, N, etc. among tents and substrate, and #3 NEC and NCP among tents and substrates.

✓We have added 4 subheadings accordingly.

*3.1 Ambient conditions*

*3.2 Water exchange quantification*

*3.2 Model output*

*3.3 Estimated biogeochemical processes*

*3.4 Incubation comparison*

Ln 215-218: How did these leakages impact incubations? Given the exchange rates were greater in these two tents, is it possible that their leaking confounded the results from these tents? How was this accounted for?

✓We checked for correlation between processes and the amount of leaking using the Kendall rank correlation test. No significant correlation was found. Text was added in the methods l. 245:

*To evaluate if water exchange rate had an impact on estimated processes, the non-parametric Kendall rank correlation test was performed. All inferred biogeochemical process rates (mineralisation, primary production, NCP, NCC, nitrification and denitrification) were tested against incubation water exchange rates.*

and the results l.305:

*The Kendall rank correlation test did not reveal significant correlation between water exchange rates and rates of mineralisation (p=0.79, tau=0.04), primary production (p=0.47, tau=0.12), NCP (p=0.75, tau=0.05), NCC (p=0.17, tau=0.21). Nitrification (p=0.81, tau=0.04), and denitrification (p=0.27, tau=0.18). The Kendall correlation coefficient tau is closer to zero than 1 in all cases, implying there is no significant association between the two tested variables.*

Ln 227, 228, 231, etc: Again, I would stick to past tense in results section, e.g. "NCP showed a clear diurnal pattern"

✓We changed the tense accordingly.

Figure 5: This figure looks like a copy-paste model output. Minimum, panels should be better labelled. However, a more informative figure could be produced that summarises the model outputs for the five substrates, day and night.

✓Figure 5 now presents model outputs for the five substrates, day and night. The old Figure 5 was made to give an illustrative view on the model output and what parameters were involved. We therefore placed it in the supplements (now Figure S1).

Figure 6: Panels should be labelled with A, B, C and D. Also, it now seems that Table 1 is redundant given this information is provided here in Figure 6. It is much easier to view as a figure, so I suggest deleting Table 1

✓Figure 6 has been labelled accordingly and table 1 has been deleted.

On this note, it would be nice to see other seawater chemistry data (AT, CT, pH, etc) presented like this in a separate figure or table. I feel the results are short and overlook baseline measurements of seawater chemistry among tents and substrate types. How did AT, CT, pH vary among substrates and across incubations?

✓All measurements are now depicted in Figure 5 but the data was centred to illustrate the different changes between incubations. A new table has been created showing all measured variables throughout incubation, day and night. It can be found in the supplements, Table S2. Text has been added l.259:

*Average ambient AT, DIC, pH, NH4 and NO3 was 2386.8 ± 13.9, 2125.5 ± 20.0, 7.9 ± 0.003, 0.31 ± 0.15 and 0.32 ± 0.14 µmol kg-1 respectively. Measured data for each incubation, inside and outside the tent for all three time-points, as well as the differences between $T_0$ and $T_4$ are presented in Fig. S2.*

Ln 242: "per" should be capitalised in PERMANOVA

✓We removed the PERMANOVA analysis.

**Discussion**

Ln 253: This is a nice contrast, but make it clear by stating the ranges of NCC you found, as well as that commonly found in the literature.

✓ This is the part of the discussion summarizing findings. More detailed ranges are stated l. 370 and onwards.

Ln 254: What does "no real gain in primary habitat" mean? Do you mean reef accretion / coral growth? If so, use this very carefully, as NCC and accretion are not always coupled. Low (or high) rates of NCC do not always correspond to low (or high) accretion rates. Did you measure primary habitat gain somehow, or is this assumed from NCC rates?

✓ We assume gain in primary habitat from NCC rates. Here we wanted to make clear what these processes translate to in terms of function, but the reviewer makes a relevant remark, we have rephrased this sentence to moderate this statement. (l. 336)

*Very low or negative NCC rates were recorded on all substrates indicative of reduced net accretion potential.*

Ln 255: As above, saying "accumulation of biomass through photosynthesis is low" may not be true.

✓This was an attempt to translate biogeochemical processes into functions to make it clear why we investigate these. We have now changed the sentence l. 337 to:

*'Net production was also low, likely indicating limited accumulation of biomass, while heterotrophic….'*

Also, use past tense throughout: "was" not "is", "were" not "are"

✓We changed the tense accordingly.

Ln 260: I am not convinced by this statement. Fig 6 shows differences among substrates, which more explicit analyses may reveal. Functional homogenisation is quite a loaded term to use from n=2-3 replicates.

✓We have revisited our analysis as advised by reviewer and removed the overall mentioning of functional homogenisation.

Ln 267, 280, 298, etc: Consider a new paper that addresses calcification and dissolution on dead and live coral surfaces: - Romanó de Orte, M., Koweek, D. A., Cyronak, T., Takeshita, Y., Griffin, A., Wolfe, K., ... & Caldeira, K. (2021). Unexpected role of communities colonizing dead coral substrate in the calcification of coral reefs. Limnology and Oceanography

✓ We thank the reviewer for this very relevant paper (with comparable results). We added it to our discussion l. 454.

Ln 269: In reference to my comment above, how was "accretion rate" calculated?

✓This sentence was removed.

Ln 274: "an" should be "and"

✓This was changed accordingly.

Ln 275: "Koweet" should be "Koweek"

✓This was changed this accordingly.

Ln 290-293: This text is useful but has no clear point as currently expressed. I also see its relevance at Ln 365.

✓We rephrased this sentence l.482 to improve clarity and placed it at the end of the discussion as suggested by the reviewer. It now reads:

*Additionally, average ambient pH at the current study site was 7.9 which is lower than average 'summer' pH, usually between 8.1 and 8.2 (den Haan et al. 2016). This may suggest that depressed calcification rates in the Piscadera Bay are indeed linked to seasonality. However, further research and additional incubations are needed to better understand the seasonal component of reef functions.*

Ln 341: As above, this key message may not be correct given the low sample size and unclear PERMANOVA analysis. The narrative and analyses must be readdressed.

✓see above

Ln 345: This level of 34-36% coral cover is very high for the Caribbean. Why was this done? Were corals intentionally moved into tents to create this high coral cover? If so, were they left to stabilize for several days after relocation? Were any metrics of coral condition measured before (and/or after) incubations? Is it possible that the corals were stressed and under-performing?

✓Corals were mot moved inside tents, we did add some substrate with bioeroding sponges or turf. But never corals. No metric of coral condition was performed, we inspected them visually, they looked healthy before and after incubations. No release of mucus was observed and coloration still present. Our results suggest that the corals in this site are indeed impacted by the organic matter overload in this area.

Referee #2 (Comments to the Author):

Webb et al., measured the community metabolism of small areas of a degraded Caribbean coral reef through in-situ incubations of benthic communities. Five incubation tents were deployed over coral, algae, and sand dominated benthos, representative of different states of coral reef degradation. Biogeochemical parameters were measured over 4-hour incubations at night and day. An inverse modelling approach was applied to the collected data. The key results were interpreted in the context of ecological function. Calcification and productivity were low and night-time respiration outweighs daytime productivity. The manuscript presents a unique and interesting approach to quantifying differences in biogeochemical processes on degraded coral reefs, however, there are some limitations to the study which should be addressed, and the inferences/conclusions that the authors make may need to be re-framed accordingly. The experimental design had low replication, and the measurements were made

at one single location over just a few days / nights. The tents were leaking during the incubations, which would also have impacted the measurements. The logistics of such in-situ incubations are very challenging, and it was a good idea deploy the tents in duplicates / triplicates, but there is some variability within substrate replicates (in terms of composition and biogeochemical activity) to suggest that they could be evaluated individually.  I think that the authors could provide some more information about the inverse modelling approach they use, and the advantages of using such an approach.

We thank the reviewer for this relevant remark. The modelling approach used in this study to estimate biogeochemical processes was first and foremost to relate the measured change in concentrations of variables ($A_T$, DIC, etc.) to the responsible metabolic processes. The reviewer is correct in pointing out that the limited number of replicates may be insufficient to permit the use of a PERMNOVA to analyse our data. Although we group the incubations in five different categories based on the dominant group, we do acknowledge that these communities do harbour differences in terms of composition and therefore refrain from performing analysis using the community composition as a categorical factor.

In line with the reviewer's key concerns:

- o   we removed the claim of functional homogenisation
- o   altered the title to put a bit of emphasise on the methodology aspect of the paper, it now reads:

'*Quantifying functional consequences of habitat degradation on a Caribbean coral reef*'

- o   We removed the PERMANOVA analysis (see below) and replaed it by PCA was conducted on a centred multivariate data set consisting of the four main biogeochemical processes (i.e., NCC, NCP, nitrification and denitrification).
- o   We rephrased some parts of the introduction and discussion and included a more cautious interpretation of the results.
- o   Although for visual ease, we do group the incubations in 5 different categories (with various colours), we refrained from performing analysis using composition as a categorical factor, rather treating each incubation as an individual.
- o   The water exchange rate was accounted for in his study by quantifying it using saline water and by measuring ambient variables (outside tent) in order to account for the water characteristics leaking in.

Specific points to highlight:

**Introduction:** Overall, the introduction is nicely written. An explanation and/or justification of the inverse modelling approach could be described either at the end of the introduction or within the methods.

We have added some text within the methods to better explain the inverse modelling approach l. 184.

*The use of inverse modelling is advantageous as it enables us to derive unknown parameters (here rates of biogeochemical processes) simultaneously from all measured data. The mathematical "state" of the incubation's dynamic system can be described based on the mass balance between $A_T$, DIC, $O_2$, $NH_4$ and $NO_3$ which is influenced by various biogeochemical processes. The rate of these processes are the unknown parameters that need to be quantified by fitting against an incomplete data set (only three-time points for $A_T$, DIC, $NH_4$ and $NO_3$).*

We also added rewrote the last paragraph of the introduction l.95 to clarify our methods.

*To account for the simultaneous convoluted influence that various processes have on measured variables, the change in their concentrations is related to the responsible metabolic processes by solving a model consisting of ordinary differential equations describing the contribution of each process to the measured chemical fluxes. With this approach, model parameters (i.e., rates of biogeochemical processes) are derived from concurrent changes in all measured variables. The aim being to provide accurate estimates of biogeochemical processes that underlie functions of the newly configured shallow Caribbean reefs.*

L38: 'similarity' does not describe species homogenisation well. Maybe rephrase.

We rephased this sentence l.43 to:

*Communities within ecosystems and across spatial scales have become more biologically homogeneous.*

L39: 'This is worrisome...'; change this to something less emotive. E.g., This threatens ….

We removed this phrasing; the sentence now reads l.43:

*Communities within ecosystems and across spatial scales have become more biologically homogeneous (Burman et al. 2012; Cramer et al. 2021) which may lead to a decrease in functional diversity therefore limiting services provided by biological communities (Matsuzaki et al. 2013; White et al. 2018).*

L55 'compromised' should be compromise; change 'was' to 'were'.

We changed these accordingly.

L59 'mirroring' might not be the correct word, as the decrease in corals is the opposite of increases in the other functional groups.

Now l.67, we changed this wording to:

*have increased alongside to the decrease in stony corals*

**Methods:** The benthic incubation tent has a great design; there are some really strong features, such as the long sampling tube which samples from the middle of the incubated area rather than a typical sampling port. The photo in figure 1 shows that a metal chain was used as a weight to hold the chamber in place, which should be mentioned in the text, and perhaps some discussion of this as a source of the reported leakage.

We thank the reviewer for his positive comment. We have added a sentence explaining we used metal chains l.114. The leakage was indeed taken into account when calculating the fluxes. The equations relating changes in concentration of the measured variables to the responsible process incorporate the leak rate as well as the ambient measurement that were always taken at the same time as interior measurements.

There are some details in this section which could be clarified to assist the reader in fully understanding how the study was carried out and to improve replicability of the experiment. In particular, the sampling regime should be further detailed: at what time of day were the tents deployed? Over how many days? Were the days and nights within the same 24-hours?

We have added text to clarify these points l.126:

*The incubations were carried out one at a time, over the study period and lasted four hours each. Prior to each incubation, the tent was place with flaps open over the substrate and lefts for a minimum of 3 hours before the incubation was started. When day incubations were terminated, the tent was left in place with flaps open until the night incubation was carried out on the same substrate. All daytime incubations were started at 10:00 and all night-time incubations were started at 18:30.*

Incubations were carried between February 12th and March 22th 2018. This information can be found l.104.

Section 2. 3: How were the compositions measured? Were they 2D only, and could a 3D area be approximated from the data you collected? Normalising the rates to substrate-specific surface area and volume might give a more accurate representation of the processes being quantified.

We measured surface areas in situ and subsequent processing with ImageJ. Relating processes to particular components within each community was not possible. The aim of this work is to look at community biogeochemical processes as a whole.

In section 2.6 it was not immediately clear if the testing of the chamber was conducted at the same time as the incubations and, if so, for how many of them? How was the impact of high salinity controlled for? A rapid change in salinity might affect coral photosynthesis for example.

We injected high saline water at the start of every single incubation to determine the water exchange rate in every tent. Although we saturated the water with salt prior to injection, this did not increase the overall salinity of the incubation by more than 1 unit each time (most of the time less than that). At this site, salinity change over the day is higher than what occurred in these tents. We therefore do not expect a significant impact on community metabolism from this increase.

We clarified when high saline water was injected in the tents l. 173.

*After sampling water at T0 for AT, CT and nutrients in each incubation, 450ml of salt-saturated water was injected into the tent.*

**Results:** In terms of structure, the order of findings could be adapted so that the key finding is first or have a summary which outlines the most important results before going through each finding in detail.

We have added the below subsections to improve clarity.

*3.1 Ambient conditions*

*3.2 Water exchange quantification*

*3.3 Model output*

*3.4 Estimated biogeochemical processes*

*3.5 Incubation comparison*

Verification of the chamber method could be its own sub-section. The rate of water exchange is high and this should be addressed in more detail. If salinity returns to the ambient level after 1-2hrs, this indicates that changes to other measured parameters are only reliable within the same time frame (i.e., the water within the tent is renewing over the course of the incubation).

Yes this is correct, this is why most measured variable and modelled behaviour level out after 2 hours. Rates are measured from the initial slope of rate of change.

We checked for correlation between processes and the amount of leaking using the Kendall rank correlation test. No significant correlation was found. Text was added in the methods l. 245:

*To evaluate if water exchange rate had an impact on estimated processes, the non-parametric Kendall rank correlation test was performed. All inferred biogeochemical process rates (mineralisation, primary production, NCP, NCC, nitrification and denitrification) were tested against incubation water exchange rates.*

and the results l.305:

*The Kendall rank correlation test did not reveal significant correlation between water exchange rates and rates of mineralisation (p=0.79, tau=0.04), primary production (p=0.47, tau=0.12), NCP (p=0.75, tau=0.05), NCC (p=0.17, tau=0.21). Nitrification (p=0.81, tau=0.04), and denitrification (p=0.27, tau=0.18). The Kendall correlation coefficient tau is closer to zero than 1 in all cases, implying there is no significant association between the two tested variables.*

It would be interesting to see TA: DIC plots and relationships between photosynthesis and calcification. If PAR data is available, you might also plot the rates against light as this could explain some of the variability. For example, if the weather conditions changed and light was reduced during some of the incubations this might explain the non-linear change in DO seen in some of the individual plots in figure S1.

The non-linear change in DO is indeed linked to light. However, as the model is unable to predict irregular oxygen evolution caused by light variability during the day-time, we do not plot that data.

We added a NCC:NCP plot to assess the position of each community within the different quadrants of the NCC vs. NCP diagram (Figure 7).

For the modelling approach presented in Fig 5 and Fig S1, the models were fit to each incubation rather than each substrate type (as they were for Fig.6). Was there a reason for this? Would it be more appropriate to describe the incubated substrates as individual substrate types rather than grouping them into categories (as per Table S1)?

Although we group them in categories, they are indeed different communities, therefore we do not fit the model through 3 incubations at a time for day data and 2 incubations at a time for night data. In line with reviewer 1 and 2's comments we have removed the PERMANOVA.

Instead, we performed a PCA (now figure 7) based on the four main biogeochemical processes estimated for each incubation. We added text to the methods l. 248:

*Principal component analysis (PCA) was used to identify grouping among the 23 tent incubations (day n = 13, night: n = 10) in relation to their biogeochemical signature (i.e., NCC, NCP, nitrification and denitrification). PCA was conducted on a centred multivariate data set consisting of the four main biogeochemical processes (i.e., NCC, NCP, nitrification and denitrification).*

And in the results l. 312:

*The PCA based on the four main biogeochemical processes revealed two main different groups between night incubations and day incubations (Figure 7A). Sand incubations were the exception as night and day incubations grouped relatively close to each other. The first two principal component axes (PC1 and PC2) explained 88.68% of the total variability within the data. PC1 described a gradient in NCP and NCC from high (negative PCA scores) to low (positive PCA scores) and an opposite pattern for nitrification and denitrification. PC2 further explains the variability in NCC and nitrification, and to a lesser extent NCP and denitrification. One of the communities dominated by bioeroding sponges (rep.1) is separate from other communities both during the day and during the night due to high rates of nitrification and denitrification compared to other communities.*

Table S1 presents useful information about the species composition of each chamber and could be used to describe reef patch as a distinct substrate type. This could be used to align individual compositions with variations in rates as the different species compositions might explain some of the variation between replicates.

We added this table for transparency about the composition of our incubations, however the aim of this research was to investigate communities as a whole. There is unfortunately no way we can relate fluxes to specific components inside the tent.

Figure S1: The data presented in these plots should be included in the manuscript. You might consider combining the plots showing the distinct replicates / substrate types with different colours or symbols. The data could be converted to rates before plotting unless there was a reason not to do this (which might also be good to explain). Figure S1 demonstrates that the DO slopes are variable for corals, with irregularities in oxygen evolution. If PAR data is available, the DO could be plotted with PAR to identify if there were changes in light to cause this (or were they all deployed on the same day?).

Figure 5 now illustrates all incubation model output.

This reviewer comment is probably based on our unclear description of the approach we used. We have therefore rephrased part of the methods (see above). We assume rates to be constant over time and therefore cannot plot them in a similar fashion to figure 5. The estimated rates are plotted out in Figure 6.

Light was indeed the cause for irregular DO evolution. The model enables predicting irregular evolution caused by light variability during the day-time. For this reason, the overall fit is usually better on night data.

Figure 4: were the tents deployed under the same conditions, or are these from distinct days?

Incubations were carried between February 12th and March 22th 2018. This information can be found l.104.

Figure 5: Plots should be combined so that all data can be presented. Visual notes: The parameter boxes should be tables to make it easier to read. All the axis tick labels should be horizonal.

Figure 5 was replotted to depict model output for all incubations. The old Figure 5 is now in Supplements (Fig. S1) with proposed alterations.

Figure 6: for each of the bars sample size should be displayed. Since n is low (either 2 or 3), the results could be displayed differently to show each data point, e.g., jitter plot, or scatterplot by assigning a numeric value to each category (e.g., coral dominated would be 5, sand would be 0). Also it looks like there would be some differences between night respiration for example, however 'no significant differences were found' in L243. Could this be due to the way the stats were run rather than a true representation?

We thank the reviewer for this relevant comment and changed the plot to a scatter plot depicting raw data and means.

In line with above comments, we removed the PERMANOVA analysis and we do not perform analysis using composition as a categorical factor, rather treating each incubation as an individual.

**Discussion:** The discussion is interesting and very well-written; however, it will likely need revisions according to the suggested edits in the results section. It would be good to include some discussion of relationships between the measured parameters (i.e., TA:DIC). Additionally, more in-depth discussion could be included to address the limitations of the study: (1) low replication and modelling with so few data points, and (2) the leakage of the chambers. The interpretation of findings should account for these uncertainties.

✓We thank the reviewer for the overall positive feedback, and we agree that addressing the limitations of our work is an essential part of the discussion. We have added the paragraph below l.358 to point out methodology considerations:

*Nonetheless, due the limited number of incubations that were carried out for this study, we interpret results with caution. Additionally, incubations were only deployed on the reef flat of one degraded reef, future application of this or similar incubation methods should consider multiple sites. Lastly, methods would be further improved by continuous monitoring of the exchange rate, rather than assuming it to be constant throughout the incubation.*

In the conclusions, information is presented about the local distribution of some coral species. This information should be described elsewhere and be incorporated into the discussion at an earlier point.

L218: if salinity returns to normal after 1-2 hours, this would indicate that the first hour or two of dDO or dTA is also lost?

We take the T0 samples before adding high saline water and correct TA if salinity has not return to ambient by T2. As mentioned previously the increase of salinity is very modest and well withing what these communities are used to.

L220: 'relatively good fit', this should be detailed further. The authors refer to Figure S1 as evidence of this, however, the model fit cannot be evaluated from figure S1 only.

This is an error; we should have also added a reference to Table S1 which presents standard errors and $p$ values for the estimated parameters. This has been corrected.

L21: potential reasons for 'irregular oxygen evolution during the daytime'?

This is due to light variability. We have added the underlined text below for clarity l.279:

"... *irregular oxygen evolution caused by light variability during the day-time*"

L254: 'no significant gain in primary habitat' is confusing because the measurements were over a matter of hours not months, so we would not expect any change to the habitat through accretion.

This is true, here we want to make clear what these processes translate to in terms of function. But as our incubations lasted only 4 hours we should be more cautious in the way we write this. We have rephased this sentence l.336 to:

*Very low or negative NCC rates were recorded on all substrates, suggesting reduced net accretion potential.*

L275: 'an' = and

Change has been made accordingly.

---

## Author Response (AR2)

The authors have done a great job revising their manuscript. The models used are more clear, and methods and analyses better-explained. Additions of relevant literature are also helpful. I have a few minor comments that should be addressed before publication.

Ln 303: suggest removing "dire"
We have removed the word "dire"

Ln 315: I suggest moving the '4.1 Method considerations' section towards the end of the discussion (perhaps before conclusion) so not to front-load the limitations of the study
We agree with the reviewer and have moved this section to the end of the discussion ln 424.

Ln 333-335: This sentence should be rephrased to avoid double-negative: "not uncommon" – something like: "Diel shifts between net calc and net diss are typical of coral reefs (Yates and Halley… etc)…"
We rephased this sentence to avoid the double negative. It now reads ln 333-335: "*Diel shifts between net calcification and net dissolution are usual for coral reefs and have been recorded on healthier systems than the one studied here."*

Ln 367: space needed between "by" and "17%"
We added a space ln 367

Ln 406: The 4.5 subheading could be clearer. Perhaps flip the text to avoid awkward use of "-" , e.g. "Similar biogeochemical processes by reef communities dominated by corals, sponges and cyanobacteria"
The subheading was changed to "Similar biogeochemical processes by reef communities dominated by corals, sponges and cyanobacteria" ln 406.

Ln 413-415: This sentence is circular "…work by Romano found comparable results….were comparable"… and needs rephrasing. E.g. "Recent work by Romano… found comparable results, showing similar daytime calc rates between live and dead coral surfaces"
We made changes accordingly ln 413-415 and the sentence now reads: "*Recent work by Romanó de Orte et al. (2021) found comparable results showing similar daytime calcification rates for live coral and dead coral substrate"*

Ln 441: Change "that" to "which"
Changes were made accordingly

Ln 143: "species composition shifts" may be better called "benthic composition shifts" or "community shifts" as the study focus seems more about general benthic cover than species-specific dominance
We agree with the reviewer and have changed the wording to "benthic composition shifts".

Ln 444: This sentence is quite unclear. What exactly does "remaining corals" mean here? The robust/resilient species that persist habitat degradation? I also caution use of "specialist" to define *Acropora*. They are more often termed 'fast-growing' or 'weedy' species.
We have altered the sentence ln 444 to : "Currently prevalent corals, although more resilient, calcify at a slower pace than previously abundant species (such as *Acropora* spp.) and cannot balance out heterotrophic processes from other functional groups."

Ln 451: Is this really "surprising" given that the sentences prior state these are "generalist" and "tolerant" corals. It would be expected that tolerant corals are more abundant in areas that have high local stress.
We removed the word surprisingly.

The conclusion should finish with 1-2 strong sentences to push home the findings of this study. Currently, the conclusion focuses on new information on tolerant species from a different paper (de Bakker et al. 2019). Stronger links to findings in this new study are required. The final sentence here essentially says that "even severely degraded reefs can regain functions", which could be misinterpreted. This study actually found that "Very low or negative NCC rates were recorded on all substrates, suggesting reduced net accretion potential." Perhaps a final sentence on the importance of protecting reefs to enhance functioning and resilience is needed.

Thank you for this relevant remark, we added these following sentences at the end of the conclusion ln. 452: "*While these sites may provide a spark of hope with regards to recovery potential, most of the reefs in the Caribbean presently reside in ecological states that closely resemble the reef site studied here (or are expected to do so in the near future). Ultimately, Caribbean reefs will benefit most notably from adequate mitigation strategies to give these systems a chance to adapt and restore key functions in the face of exacerbating environmental conditions.*"